# Unconventional interfacial water structure of highly concentrated aqueous electrolytes at negative electrode polarizations

Chao-Yu Li [1,5], Ming Chen [2,5], Shuai Liu [3], Xinyao Lu[4], Jinhui Meng[1], Jiawei Yan [3], Héctor D. Abruña [4], Guang Feng [2] ✉ & Tianquan Lian [1] ✉

Water-in-salt electrolytes are an appealing option for future electrochemical energy storage devices due to their safety and low toxicity. However, the physicochemical interactions occurring at the interface between the electrode and the water-in-salt electrolyte are not yet fully understood. Here, via in situ Raman spectroscopy and molecular dynamics simulations, we investigate the electrical double-layer structure occurring at the interface between a water-in-salt electrolyte and an Au(111) electrode. We demonstrate that most interfacial water molecules are bound with lithium ions and have zero, one, or two hydrogen bonds to feature three hydroxyl stretching bands. Moreover, the accumulation of lithium ions on the electrode surface at large negative polarizations reduces the interfacial field to induce an unusual "hydrogen-up" structure of interfacial water and blue shift of the hydroxyl stretching frequencies. These physicochemical behaviours are quantitatively different from aqueous electrolyte solutions with lower concentrations. This atomistic understanding of the double-layer structure provides key insights for designing future aqueous electrolytes for electrochemical energy storage devices.

With the rapidly increasing demand for renewable energy and mobile devices, the development of next-generation electrochemical energy storage systems has become a key objective in energy research[1,2]. In electrochemical energy devices, the electrolyte, sandwiched between the positive and negative electrodes, is an essential component determining the performance. Aqueous electrolytes have safety advantages over aprotic organic electrolytes, such as non-flammability and low toxicity[3–6]. However, conventional aqueous electrolytes are limited by their narrow electrochemical window of 1.23 V due to the hydrogen/oxygen evolution reactions[7]. This limitation can be overcome in highly concentrated aqueous electrolytes, also known as water-in-salt (WiS) electrolytes[8], which have been shown to significantly widen the electrochemical stability window to

over 3.0 V, delivering enhanced performance in Li-ion batteries[8,9], supercapacitors[10], and $CO_2$ reduction reactions[11].

Recent reports on WiS electrolytes have drawn considerable attention to their bulk properties. For example, in 21 molal (m, moles of salt per kg of water) lithium bis(trifluoromethane sulphonyl) imide (LiTFSI) WiS electrolytes, more than 90% of the total mole of water molecules are bound with Li[+], and both the strong interaction with Li[+] and small free water concentration (<10% of the total mole of water) have been suggested to contribute to the enhanced electrochemical stability of WiS electrolytes[8,12–14]. Furthermore, water molecules form nanoscale channels to facilitate Li[+] transport despite the large macroscopic viscosity of WiS electrolytes[15]. In addition to bulk properties, the performance of the electrochemical device also depends critically

[1]Department of Chemistry, Emory University, Atlanta, GA 30322, USA. [2]State Key Laboratory of Coal Combustion, School of Energy and Power Engineering, Huazhong University of Science and Technology (HUST), Wuhan 430074, China. [3]State Key Laboratory of Physical Chemistry of Solid Surfaces, College of Chemistry and Chemical Engineering, Xiamen University, Xiamen 361005, China. [4]Department of Chemistry and Chemical Biology, Cornell University, Ithaca, NY 14853, USA. [5]These authors contributed equally: Chao-Yu Li, Ming Chen. ✉e-mail: gfeng@hust.edu.cn; tlian@emory.edu

on the electrical double-layer (EDL) structure at the electrode–electrolyte interface, a subnanometer region where electrochemical transformations occur. Molecular dynamics (MD) simulations suggest that the high salt concentration in WiS electrolytes significantly alters the $Li^+$ solvation and EDL structures compared to low-concentration electrolytes[16–18]. However, such modeling results have yet to be supported by detailed experimental characterization of the EDL structure at the electrode/WiS electrolyte interface. Previous studies of low-concentration electrolytes have shown that a thorough characterization of EDL structures is crucial for the mechanistic understanding of device performance[19–27]. Extending such studies to the EDL of WiS electrolytes, especially in high electrode polarization regions (e.g., potentials negative than the onset of hydrogen evolution reaction), remains challenging[13,28].

Herein, the potential-dependent EDL structure of highly concentrated LiTFSI aqueous electrolytes on an Au(111) electrode was studied at electrode polarizations varying from 0 to −1.55 V (vs. potential of zero charge, PZC). Comparing the potential-dependent vibrational spectra of interfacial water and ions obtained by in situ electrochemical shell-isolated nanoparticle-enhanced Raman spectroscopy (SHINERS)[22,29] with detailed microstructures obtained by MD simulations provides atomistic insights into the EDL structure in WiS electrolytes. The comparison reveals that the interfacial water molecules show three OH stretching bands due to their different hydrogen-bonded structures dictated by interactions among the water, $Li^+$, and $TFSI^-$. The accumulation of $Li^+$ at the electrode surface at high negative polarizations leads to an unusual hydrogen-up (H-up) interfacial water structure that is not observed in dilute aqueous electrolytes and reduces the electric field strength on the interfacial water to give rise to an unexpected blue shift of the OH stretching vibration.

## Results

### In situ probing interfacial water in highly concentrated aqueous electrolyte

As illustrated in Fig. 1a, to study the EDL structure, an electrochemical SHINERS method was used (details in Supplementary Fig. 1 and Supplementary Information), which has been proven to be suitable for the investigations at the electrochemical interface, such as specific adsorption of sulfate ion[30], pyridine[31], and hydrogen[29], and most importantly and relevantly, enables an in situ molecular-level probe of the structures of electrical double layer and interfacial water on single-crystal electrode surfaces[22,29,32]. The Raman signal of the interfacial layer is significantly enhanced in the junction between the Au(111) surface and the core/shell $Au/SiO_2$ nanoparticles due to the enhanced optical field strength. The thin $SiO_2$ layer (2 nm in thickness) is electrochemically inert, which can insulate the Au particles (~57 nm in diameter) from the single-crystal Au electrode while still provides enhancement of the electric field (see Supplementary Fig. 1 for the TEM characterization of Au nanoparticles and their size distribution histogram). MD simulations, as shown in the schematic in Fig. 1b and with details described in the Method section, were utilized to investigate the atomistic structure of the EDL and the water hydrogen-bonded structure at the Au(111) electrode-WiS electrolyte interface.

The vibrational spectra of interfacial water in the OH stretching region and their changes with applied potentials were measured by in situ electrochemical Raman spectroscopy in 21 m LiTFSI aqueous electrolytes. All electrode potentials are relative to the PZC of the Au(111) electrode, which is around −0.1 V vs. Ag/AgCl electrode (see Supplementary Figs. 2, 3 for corresponding cyclic voltammogram and PZC measurement of Au(111) in 21 m LiTFSI aqueous electrolyte). The Raman spectra at +0.5 and +0.9 V shows negligible dependence on the applied bias (Supplementary Fig. 4) and agree well with bulk water Raman spectra of WiS electrolyte. Therefore, the spectrum at +0.9 V was subtracted from the total Raman signal to reveal the spectra of interfacial water. Shown in Fig. 2a are the Raman spectra of the interfacial water OH stretching mode in the 3200–3600 $cm^{-1}$ region from +0.5 to −1.55 V. Inspired by the assignments of Raman spectra of water in the low-concentrated aqueous electrolytes[22,32–36], the spectra can be well fitted by the sum of three Gaussian bands, Peak 1, 2, and 3 (with increasing frequencies), suggesting three major types of water molecules in the EDL. As shown in Fig. 2b, c, both the frequencies and intensities of these three bands show a strong dependence on the applied potential, reflecting the bias-dependent change of the interfacial water structure and electric field[22]. The Raman frequencies of Peak 1–3 decrease linearly from +0.1 to −1.15 V, consistent with Stark effect induced frequency shifts[22,37–40], which is caused by the monotonically increasing total electric field experienced by interfacial water in the potential region of 0 to −0.96 V, as disclosed by MD simulations in Fig. 2d. However, at more negative potentials, from −1.15 to −1.55 V, the Raman frequencies of these three peaks increase, which is not observed in 7 m LiTFSI aqueous electrolyte (Supplementary Fig. 5) and have not been reported previously[22,40]. MD simulations reveal an

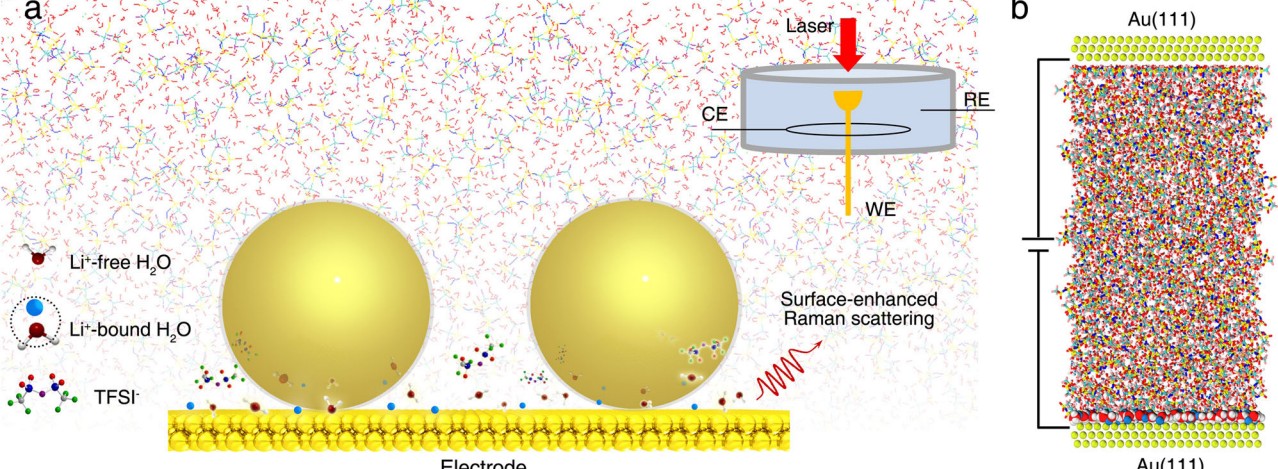

**Fig. 1 | Probing the electrochemical double-layer structure at the electrode|electrolyte interface. a** Schematic of in situ probe of the EDL using SHINERS method. Bottom left: schematic structure of the $Li^+$-free water, $Li^+$-bound water molecules, and $TFSI^-$ anion, respectively; Top right: schematic of a spectro-electrochemical cell, where an Au(111) electrode, a Pt wire, and an Ag/AgCl electrode were used as the working electrode (WE), counter electrode (CE), and reference electrode (RE), respectively. The large spheres in the bottom center are the core/shell $Au/SiO_2$ nanospheres. **b** Typical MD simulation snapshot of 21 m WiS electrolyte in contact with Au(111) electrodes under applied bias. The interfacial water molecules on the negative electrode surface are magnified.

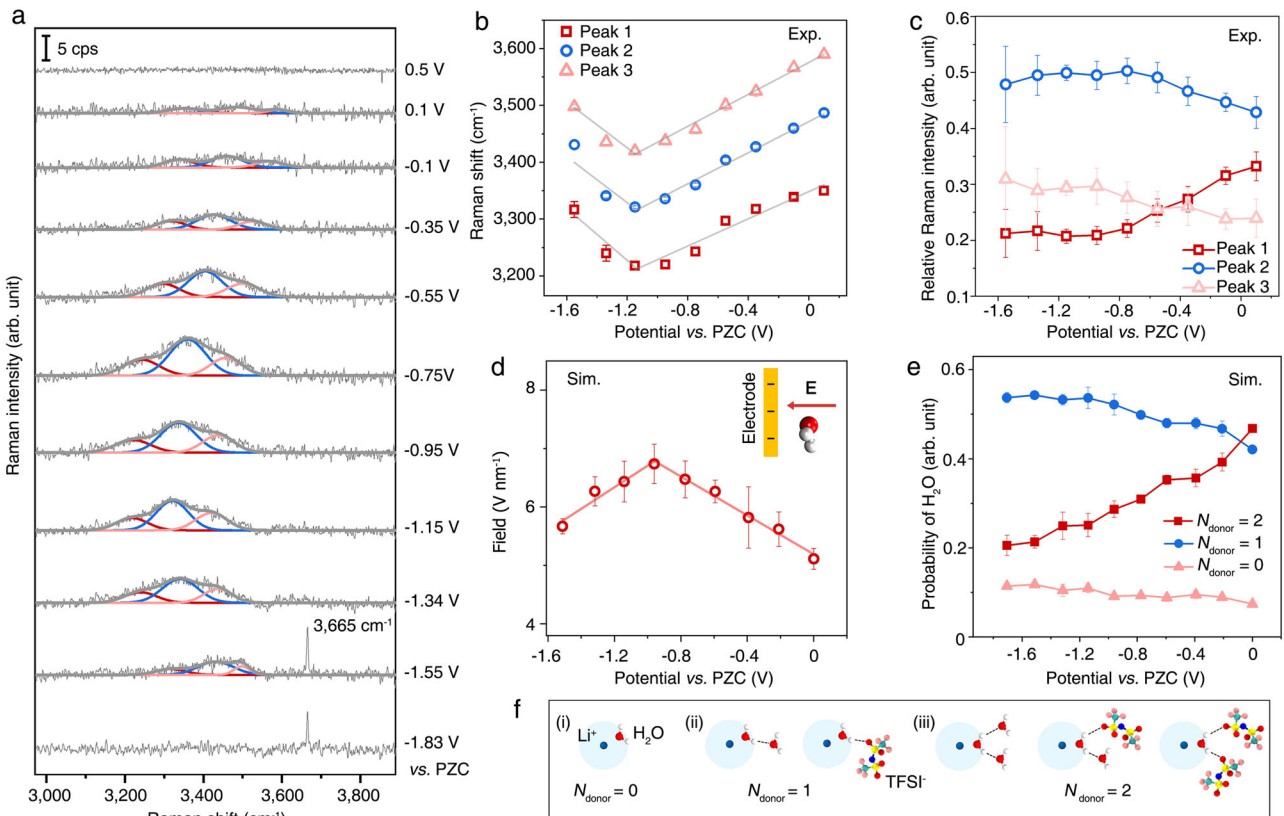

**Fig. 2 | Vibrational Raman spectra of interfacial water at Au(111)|WiS electrolyte (21 m LiTFSI) interface. a** In situ electrochemical Raman spectra of the OH stretching mode of interfacial water at the Au(111) surface measured in 21 m LiTFSI aqueous electrolyte (gray curves) and their fits to the sum of three Gaussian bands of increasing frequency: Peak 1 (red), 2 (blue), and 3 (pink). **b** Potential-dependent frequencies of Peak 1 (red squares), 2 (blue circles), and 3 (pink triangles) of interfacial water obtained from spectral fits in panel **a** and their linear fit (solid lines) in regions of +0.1 ~ −1.15 V and −1.15 ~ −1.55 V. **c**, Potential dependence of the relative intensities of Peak 1 (red squares), 2 (blue circles), and 3 (pink triangles) obtained from the fits in **a**. Herein, a relative Raman intensity profile representing the change in the portion of total intensity was used to avoid the variation of plasmonic enhancement during the potential scan. **d**, Simulated electric field (**E**) strength (red circles) experienced by the interfacial water and the linear fit (solid red line) in

regions of 0 ~ −0.96 V and −0.96 ~ −1.51 V. Inset: schematic of the interaction between the electric field and interfacial water. The direction of **E** is taken to be the direction of the force it exerts on a positively charged particle, e.g., from the positive electrode to the negative electrode. **e** Simulated potential-dependent probability of interfacial water with donor numbers of 0 (pink triangles), 1 (blue circles), and 2 (red squares), respectively. As shown in the schematic in Supplementary Fig. 11, H-bond formation was determined based on the geometric criteria of hydrogen bond length ($r_{HB} \leq 0.35$ nm) and angle ($\alpha_{HB} \leq 30°$)[17,61]. The error bars in **b**–**e** present the standard error in experiments or simulations. **f** Schematics of Li⁺-bound interfacial water with donor number changing from (i) 0 to (ii) 1 and (iii) 2, respectively. In the formation of an H-bond with a Li⁺-bound water molecule as a donor, another water molecule or TFSI⁻ can act as an acceptor.

unexpected decrease in the total electric field experienced by the interfacial water at the highly negative potential region from −0.96 to −1.51 V (Fig. 2d), which accounts for this blue shift in the water OH stretching mode[38,39]. The origin of this unexpected change in the interfacial electric field will be further discussed below.

The absolute intensities of Raman bands depend on the population of interfacial species and their Raman enhancement factors, both of which can change with potential, and their relative contributions cannot be easily separated. Thus, only the relative intensities of Raman bands and their potential dependence are analyzed, which will be compared with the potential-dependent probability of finding interfacial species of different hydrogen-bonded structures obtained from the MD simulation. The relative intensities of three OH stretching bands, defined as the ratio of the intensity of a band to the total Raman intensity of all bands, show different bias dependence, as presented in Fig. 2c. From +0.1 to −1.15 V, the relative intensity of Peak 1 decreases while those of Peak 2 and 3 increase. From −1.15 to −1.55 V, the relative intensities of all three peaks show a much smaller bias dependence. Noticeably, at −1.55 and −1.83 V, a sharp peak with a bandwidth of ~4 cm⁻¹ appears at 3665 cm⁻¹, which can be attributed to the deposition of LiOH on the Au(111) surface to form a solid electrolyte interphase (SEI) layer (Supplementary Fig. 6). This observation is in line with the

findings of a previous X-ray diffraction measurement, which shows that at negative potentials, Li ions accumulate at the electrode surface and react with OH⁻ (released from the hydrogen evolution reaction) to form LiOH precipitates[13]. This surface reaction has been suggested to passivate the electrode surface and suppress the water-reduction reaction, leading to a widened electrochemical stability window during cycling[13].

In addition, as shown by the cyclic voltammetry (CV) measurement of Au(111) in 21 m WiS electrolyte (Supplementary Fig. 2), a cathodic current starts at ~ −0.8 V, which can be attributed to the hydrogen evolution reaction and also the reduction of TFSI⁻[8]. To characterize the surface change, we held the potential at −0.8 and −1.2 V on Au electrodes for 24 h and 30 min, respectively, and then carried out ex situ Raman spectroscopic measurements to probe the electrode surface. As shown in Supplementary Fig. 7, there is no observable Raman signal of SEI (i.e., the deposition of LiOH) on the electrode surface under such potentials. Furthermore, the atomic force microscope (AFM) measurements were performed to study the surface morphology of the electrode surface after holding at about −1.2 and −1.8 V. As revealed in Supplementary Fig. 8a, b, the electrode surface is still atomically flat after holding at ~−1.2 V for 5 min, where the characteristic monatomic steps of Au(111) surface can be clearly

seen by the AFM measurement, indicating the high surface smoothness. In sharp contrast, after holding at ~−1.8 V, the surface morphology of the electrode became much more roughened with several valley-like regions (Supplementary Fig. 8c, d), suggesting the formation of SEI at this potential.

It has been well established that in aqueous solutions and dilute electrolytes, the frequency of the OH stretching mode of interfacial water is determined by the hydrogen bond (H-bond) environment of water molecules, giving rise to distinct peaks for ice-like water[33,34], liquid-like water[33,34], and dangling OH bonds[22,36]. To help assign the observed interfacial OH stretching spectra, MD simulations were performed to explore the structure of water molecules and the corresponding H-bond network at the electrode-WiS electrolyte interface. As shown in Supplementary Fig. 9, more than 93% of the total mole of interfacial water molecules are within the first solvation shell of $Li^+$ (referred to as $Li^+$-bound water), and less than 7% of the total mole of water molecules are outside the first solvation shell of $Li^+$ ($Li^+$-free water). This can be attributed to the strong interaction between $Li^+$ and water molecules under the high salt concentration[16,17]. Furthermore, the simulated number density of $Li^+$-bound water increases during the cathodic scan, likely induced by the accumulation of $Li^+$ on the electrode surface at more negative potentials. The percentage of interfacial $Li^+$-bound water molecules is much higher than that in 7 m LiTFSI aqueous electrolytes (~75% of the total mole of interfacial water molecules are within the first solvation shell of $Li^+$, see Supplementary Fig. 10).

The variation in H-bond numbers alters the structure of the EDL and can be observed by Raman frequency shifts because the frequency of the OH stretching mode ($\nu_{OH}$) shifts towards a higher region in water molecules with a lower H-bond number[35]. The H-bond formation was determined based on the geometric criteria in Supplementary Fig. 11. The potential-dependent probabilities of finding interfacial $Li^+$-bound water with different H-bond donor and acceptor numbers are calculated and summarized in Supplementary Table 1. $Li^+$-bound water molecules with zero acceptor number dominate the interfacial region, since the interfacial $Li^+$-bound water molecules, with their O atom interacting with the $Li^+$ (schematically shown on the left of Fig. 1a), cannot serve as an H-bond acceptor. Therefore, as illustrated in Fig. 2e, f, the main difference in the H-bond environment of interfacial $Li^+$-bound water molecules is determined by their H-bond donor number ($N_{donor}$), which ranges from $N_{donor} = 0$, 1, and 2. As shown in Fig. 2e, the fraction of interfacial water molecules with $N_{donor} = 2$ decreases gradually, and the proportions of water molecules with $N_{donor} = 1$ and 0 increase correspondingly at more negative potentials. On the basis of the relationship between $N_{donor}$ and $\nu_{OH}$ established in dilute electrolyte solutions[22,32–36], Peak 1, 2, and 3 (in the order of increasing wavenumbers) of the interfacial Raman spectra are attributed to interfacial $Li^+$-bound water molecules with two ($N_{donor} = 2$), one ($N_{donor} = 1$), and zero ($N_{donor} = 0$) H-bonds, respectively. The simulated potential-dependent probability of these species (Fig. 2e) agrees qualitatively with the observed trend of their relative Raman intensities (Fig. 2c), providing further support for this assignment. Although our result suggests that the well-established relationship between $N_{donor}$ and $\nu_{OH}$ of water for dilute solutions can also be applied in concentrated electrolytes, this notion should be further examined in future studies.

## Microscopic structure of EDLs in highly concentrated aqueous electrolytes

MD simulations reveal the detailed atomistic structure of interfacial water, $Li^+$, and TFSI⁻ in the EDL of WiS electrolytes. The structure of interfacial water and $Li^+$ in the 21 m LiTFSI aqueous electrolyte can be visualized by the number of density profiles of interfacial water oxygen (Fig. 3a) and hydrogen (Fig. 3b) atoms and $Li^+$ (Fig. 3c), as well as the water orientation (Fig. 3d, e), both of which show pronounced

dependences on the applied potential. The orientation of water relative to the electrode surface can be described by two angles: 1) the angle between the normal vectors to the water plane and the electrode surface ($\theta_{normal}$), shown in Figs. 2, 3d) the angle between the water dipole and surface normal ($\theta_{dipole}$), shown in Fig. 3e. Meanwhile, a differential 2D angular distribution of two OH groups of $Li^+$-bound water relative to the arrangement of water under PZC is shown in Fig. 3f.

The number density profiles of water hydrogen and oxygen atoms and $Li^+$ exhibit distinct peaks at three distances from the electrode, indicating their interfacial structural order. Specifically, a sharp peak is observed for the oxygen atoms located at the second layer (-0.26 nm to the surface), and at more negative potentials, the peak height increases, with its position shifting closer to the electrode surface (Fig. 3a). However, as shown in Fig. 3b, the distribution of hydrogen atoms shows more potential-dependent changes. At the PZC, the hydrogen atoms are located at the same layer as the oxygen atoms, with the $\theta_{normal}$ distribution exhibiting two peaks around 20 and 160° (Fig. 3d) and a $\theta_{dipole}$ distribution peaking at 105° (Fig. 3e), which indicates that the interfacial water adopts a configuration nearly parallel to the electrode surface, consistent with previous work in dilute aqueous electrolytes[22]. As the polarization increases to ~−1.0 V, water molecules get noticeably re-arranged: the H atom distribution shows a new peak at the first layer (-0.17 nm to the surface) and a decrease of the peak height at the second layer (Fig. 3a, b); the distribution of $\theta_{normal}$ becomes less ordered (Fig. 3d) and the peak of $\theta_{dipole}$ distributions shifts to 115° (Fig. 3e). These changes can be attributed to the reoriented "dipole-down" structure (i.e., the water dipole points towards the electrode surface), in accord with previous reports of water structure on gold electrodes by X-ray absorption spectroscopy[41] and Raman spectroscopy[22]. However, under high polarization, we observe that a shoulder of H atoms grows in at the third layer (-0.35 nm to the surface, see Fig. 3b), and a peak of $\theta_{dipole}$ distribution at around 60° gradually increases (Fig. 3e). Such structure could be ascribed to an unusual "dipole-up structure" (i.e., the water dipole points away from the electrode surface). Such an unusual interfacial water structure has never been reported on highly negatively charged surfaces[22,33,36,37]. In order to describe the structure transition of interfacial $Li^+$-bound water molecules with the applied potential more accurately, we then calculated the differential 2D angular distribution of two OH groups of $Li^+$-bound water relative to the arrangement of water under PZC. Specifically, the OH bond of a water molecule can be classified into "H-up", parallel, and "H-down" with the angle between the OH bond of water and the normal of the electrode surface ranging in 0–70°, 70–110°, and 110–180°, respectively (Supplementary Fig. 12). As shown in Fig. 3f, the arrangement of $Li^+$-bound water molecules adjusts from parallel to "H-down" under low polarization. Nevertheless, under high polarization, though the major $Li^+$-bound water molecules transfer from parallel to "H-down", part of the water molecules transfer into an unusual H-up configuration.

Analysis of the EDL structure suggests that the change of interfacial water orientation is induced by the variation of interfacial $Li^+$ distribution. As shown in Fig. 3c, at the PZC, most Li ions are located at the third layer (-0.35 nm to the surface), and a small amount of Li ions is located at the second layer, the plane of water (-0.26 nm to the surface). At more negative potentials, $Li^+$ accumulates in the inner region of the EDL (-0.17 nm to the surface), agreeing with our experimental observation of LiOH deposition on Au(111) at −1.55 and −1.83 V (Fig. 2a and Supplementary Fig. 6). The accumulation of $Li^+$ partially screens the electric field, enabling the H-up structure of interfacial water. As a comparison, the $Li^+$ distribution in the low concentration (7 m) LiTFSI aqueous electrolyte shows negligible density in the first atomic layer (Supplementary Fig. 13a–c), which inhibits the occurrence of the dipole-up structure of interfacial water (Supplementary Fig. 13d, e).

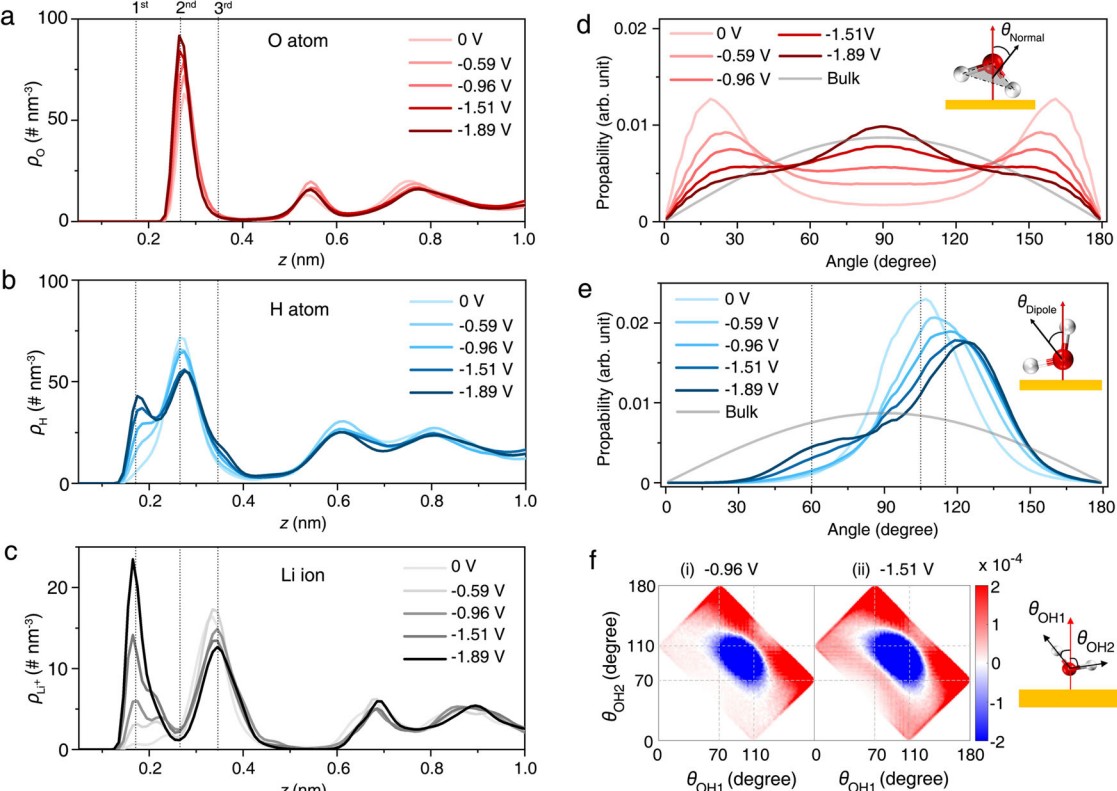

**Fig. 3 | Microscopic structure of the electrical double layer at Au(111)|WiS electrolyte (21 m LiTFSI) interface. a–c** Atom number densities ($\rho$) of oxygen (**a**) and hydrogen (**b**) of water, and Li⁺ (**c**) in 21 m LiTFSI aqueous electrolyte. **d**, **e**, Normal orientation (**d**) and dipole orientation (**e**) of interfacial water. The normal orientation is defined as the angle between the normal of the water plane and the normal of the electrode surface; the dipole orientation is the angle between the water vector and the normal of the electrode surface. The solid gray line represents the orientation of bulk water; three dash lines in **e** denote the peak at (i) 105˚, (ii) 115˚, and (iii) 60˚, respectively. **f** Differential 2D angular distributions of two OH groups of Li⁺-bound water at (i) −0.96 V and (ii) −1.51 V relative to the arrangement of water under PZC. $\theta_{OH}$ is defined as the angle between the OH bond of the water and the normal direction of the electrode surface (the corresponding schematic is shown on the right). In **d–f**, the water molecule is represented by the white and red spheres.

In addition to the structure of interfacial Li⁺ and water at negative electrode polarizations, our study also reveals the structure of the counter ion (i.e., TFSI⁻) within the interfacial region. As shown in Supplementary Fig. 14a, b, the vibrational Raman spectra exhibit multiple bands of TFSI⁻ in the 250 to 1300 cm⁻¹ spectral region. These bands show a similar lack of potential dependence, and only the sharpest peak at 746 cm⁻¹ (S-N-S bending mode, $\delta_{\text{S-N-S}}$) is discussed. Over the entire potential range (+0.5 to −1.55 V), the $\delta_{\text{S-N-S}}$ peaks show negligible frequency shifts (Supplementary Fig. 14a), while the integrated intensity changes by < ~10% over the spectral region (Supplementary Fig. 14b). The MD simulations reveal that at the PZC, the TFSI⁻ number density distribution has a prominent peak of their center of mass at 0.4 nm from the electrode surface (Supplementary Fig. 14c), indicating that TFSI⁻ ions sit above the interfacial water layer (~0.26 nm to the surface) and slightly above the Li⁺ layer (~0.35 nm to the surface). At more negative potentials, the number density of TFSI⁻ at the 0.4 nm peak decreases, suggesting that the increased negative charge of the electrode surface pushes more TFSI⁻ ions further away from the surface by electrostatic repulsion while attracting more Li⁺ into the inner layer by electrostatic attraction (Fig. 3c). Detailed analysis of the MD simulation results show that at the PZC, the hydrogen bond acceptors of interfacial water are mostly the sulfonyl oxygen of TFSI⁻ (Supplementary Fig. 15), for ~93.9% for water with one H-bond donor and ~92.7% for water with two H-bond donors. At more negative potentials, the percent of H-bonds with TFSI⁻ acceptor decreases as more TFSI⁻ ions are pushed further away from the electrode surface.

The analysis of the combined in situ electrochemical Raman spectroscopy and constant-potential MD simulations results have led us to propose an EDL model in highly concentrated aqueous electrolytes. As depicted schematically in Fig. 4, most interfacial water molecules are within the first solvation shell of Li⁺ with their O atoms interacting with the Li⁺ and their H atoms forming 0, 1, and 2 H-bonds, with TFSI⁻ as the principal H-bonding acceptor. The electrolyte in the double layer shows a structural order in the density profile along the normal to electrode surface: with water molecules closest to the electrode surface interacting with the Li⁺ and TFSI⁻ ion layers above. Under low polarization (from 0 to ~−0.6 V), the interfacial water molecules adopt parallel and H-down structures (Fig. 4a). However, under high polarization (from ~−1.5 to −1.9 V), Li ions accumulate on the electrode surface, inserting between the electrode and water layer (Fig. 4b). As a result, the amount of interfacial Li⁺-bound water adopting an "H-up" orientation increases from ~10% (PZC) to ~20% (high polarization); meanwhile, the amount of interfacial Li⁺-bound water with a parallel orientation decreases from ~43 to ~16% (Supplementary Fig. 16). Specifically, in terms of the "H-up" structure of Li⁺-bound water, ~20% of interfacial Li⁺-bound water molecules form zero H-bond (i.e., two dangling OH bonds), and ~55% of interfacial Li⁺-bound water molecules adopt one H-bond (i.e., one dangling OH bond); meanwhile, the rest ~25% of interfacial Li⁺-bound water molecules form two H-bonds (i.e., zero dangling OH bond, Supplementary Fig. 17).

## Molecular insights into the structural transition of interfacial water

As shown in Fig. 2b, d, the potential-dependent OH stretching frequency shifts of interfacial water molecules with different H-bond

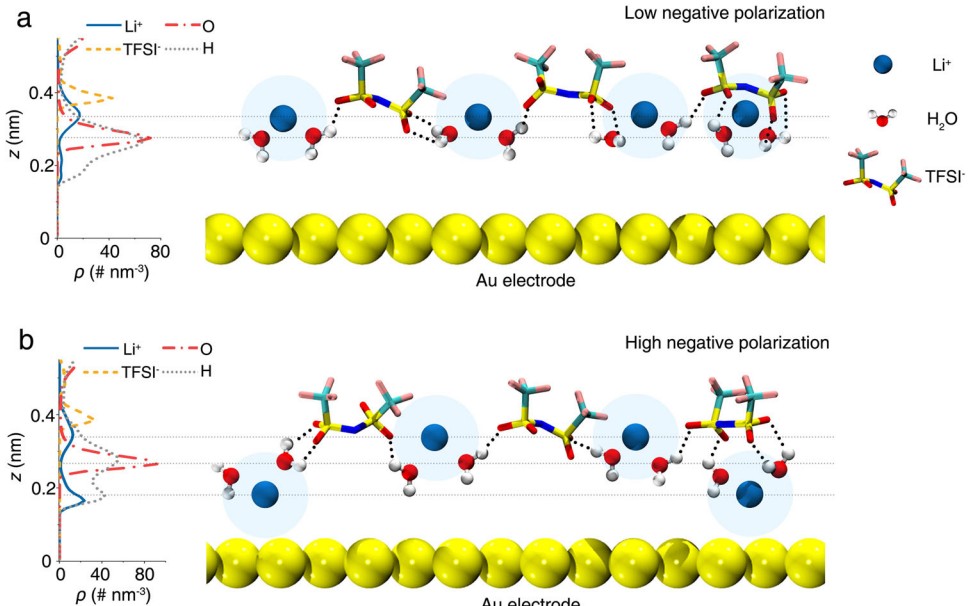

**Fig. 4 | Schematic atomistic EDL structure in highly concentrated aqueous electrolyte. a, b** Number densities profiles of various interfacial species (left), and the corresponding schematic EDL structure (right) under low negative polarization, from 0 to -−0.6 V (**a**) and high negative polarization, from -−1.5 to −1.9 V (**b**). Horizontal dashed lines indicate the peaks of density profiles of Li+, O, and H atoms of water and the center of mass of TFSI-. The deposition of LiOH at the Au(111) surface at −1.55 V is omitted.

numbers correlate well with the average electric field experienced by the interfacial water, indicative of the Stark effect-induced frequency shift[22,35,38,39]. To understand how the potential-dependent double-layer structure change leads to the observed frequency shift, we decompose the total field on the interfacial water (red circles) into contributions from the electrode (H2O-electrode, gray bar), water molecules (H2O-H2O, orange bar), and Li+ and TFSI- ions (H2O-LiTFSI, blue bar), as presented in Fig. 5a. Details can be referred to Methods. It is found that, in the 21 m electrolyte, the strength of the electric field from the electrode increases linearly with the applied potential (Fig. 5a), due to the monotonically increased surface charge density (Fig. 5b). Similarly, the field strength from the water molecules, pointing at the oppositive direction, also increases monotonically with the potential (orange bar in Fig. 5a). Consequently, the unusual transition of the total electric field results mainly from the contribution of Li+ and TFSI- ions. Specifically, TFSI- ions stay further from the electrode than the interfacial water layer (Fig. 4) and the number of TFSI- ions in the EDL region remains nearly unchanged with polarization (Supplementary Fig. 18), generating a nearly polarization-independent electric field at the opposite direction from that of the negatively charged electrode. As shown in Figs. 3c, 5c, there are two layers of Li ions: an outer layer above the interfacial water that produces an electric field having the same direction as that from the negatively charged electrode, and an inner layer between the electrode and the interfacial water layer that generates the electric field in the inverse direction (see the Schematic in Fig. 5d). The peak locations of the Li+ layers are nearly independent of polarization. The amount of Li ions in the outer layer increases slightly from 0 to −0.96 V and then decreases more noticeably from −0.96 to −1.51 V; while the amount of Li ions in the inner layer increases in the whole potential range, but the slope of increase is much larger from −0.96 to −1.51 V (Fig. 5c). The difference in the number of Li ions between the outer and inner layers ($\triangle\rho$) decreases gradually in the potential of 0 to −0.96 V, and then greatly at the potential region from −0.96 to −1.51 V. Therefore, Li+ accumulating on the electrode surface, which screens the negative electrode

surface charge, plays a dominant role in the unconventional transition of the total electric field at -−1.0 V (Fig. 5a).

For the lower concentration electrolyte (7 m LiTFSI), the reverse frequency shift of the OH stretching mode and the deposition of LiOH were not observed experimentally at high negative potentials, up to −1.75 V (Supplementary Fig. 5), although the slope of the frequency shift decreases. Unfortunately, measurements at more negative potential are difficult due to the great H2 evolution. Moreover, MD simulations show no discernable Li+ accumulation in the inner layer of the EDL (Supplementary Fig. 13c). It is possible that the interfacial field may also be screened at the negative potential range in the lower concentration electrolyte, but the extent of screening is smaller and does not lead to a blue shift of the Raman frequencies of the OH stretching bands.

Fundamentally, the unique atomic structure of the EDL in highly concentrated aqueous electrolytes can be used to reveal the nature of the water electrosorption in wet ionic liquids and also the mechanism of enhancing/expanding the voltage window stability via adding salt[8,16,17]. Recently, an "H-up" water structure was predicted to lower the potential of hydrogen evolution reaction on the negative electrode in humid ionic liquids[17]. Therefore, the unusual "H-up" structure of interfacial water and the H-bond network with TFSI- anions at very negative potentials in this work may be used to tune the onset potential of water-reduction reaction in aqueous electrolytes for electrochemical energy storage devices.

In summary, by combining in situ vibrational spectroscopy and constant-potential MD simulations, we have studied the atomistic structure of the EDL of highly concentrated (21 m LiTFSI) aqueous electrolytes at an Au(111) electrode. The Raman spectra of interfacial water OH stretching modes show three bands with potential-dependent relative intensities and frequencies. MD simulations reveal that >93% of the interfacial water molecules are in the first solvation shell of the Li+. These interfacial water molecules cannot serve as H-bond acceptors but donors because their O atoms are coordinated with the Li+. Their H-bond environments differ in the H-bond donor number, ranging from 0, 1, and 2, corresponding well

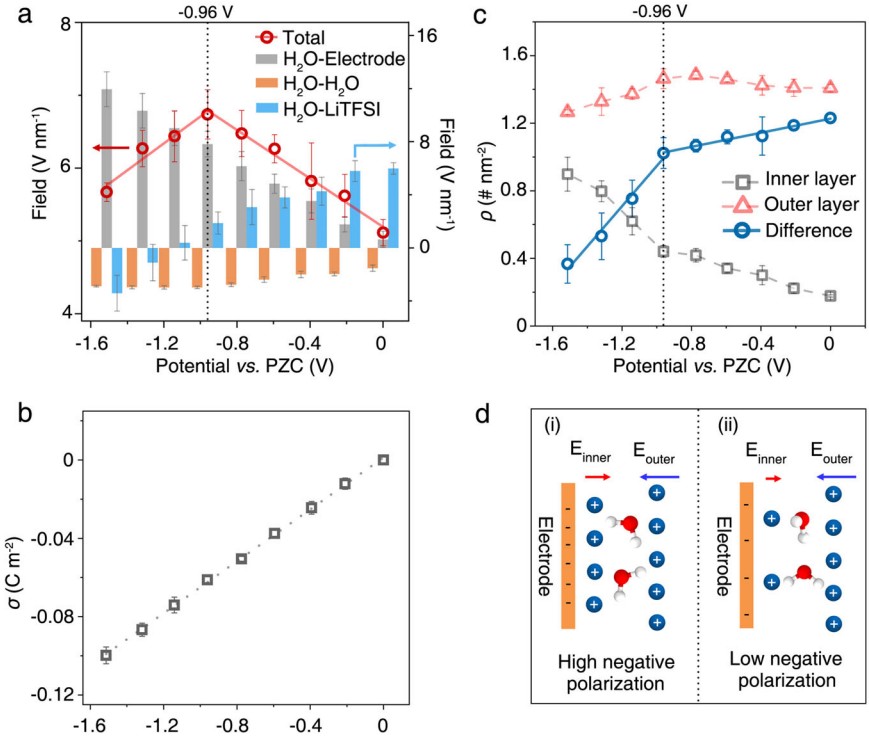

**Fig. 5 | Molecular insights into electric field change. a** Simulated electric field strength experienced by the interfacial water as a function of applied potential from 0 to −1.51 V. The total electric field (red circles) is fitted linearly (solid red line), shown on the left axis. Such total electric field is decomposed into the $H_2O$-electrode (gray bar), $H_2O$-$H_2O$ (orange bar), and $H_2O$-LiTFSI electrolyte (blue bar) interactions, respectively, which are rescaled in the right axis for clarity. **b** Surface charge density (σ) as a function of potential. The gray dot line in (**b**) is to guide the eyes. **c** Potential-dependent accumulative number density (ρ) of Li ions in the inner (gray squares) and outer (red triangles) layers, and the corresponding difference (blue circles) in the number density of ions between outer and inner layers. In **a**–**c**, the error bars present the standard error. **d** Schematic of electric field experienced by interfacial water induced and generated by Li ions in the inner (i) and outer (ii) layers. Li ions in the outer layer generate the electric field having the same direction as that generated by the negatively charged electrode; while the inner layer produces the inverse electric field. The water molecules are represented by the white and red spheres.

to the three observed OH stretching bands with decreasing frequencies. This assignment is supported by the good agreement between the potential dependence of the observed relative Raman intensities of these bands and the simulated probability of water molecules with these donor numbers.

Ascribed to Stark effect induced vibrational frequency shifts, the frequencies of all three OH bands shift to lower values at more negative electrode polarizations from the PZC to −1.15 V; while they exhibit an unexpected blue shift from −1.15 to −1.55 V, which has not been observed in low-concentration electrolytes. MD simulations reveal that over this potential range, Li ions in WiS electrolytes accumulate on the electrode surface, inserted between the electrode and the first layer of water molecules. This reduces the electric field strength experienced by the interfacial water molecules, giving rise to the observed blue shift of their OH stretching frequencies. It also leads to unusual "H-up" interfacial water molecules with the dipole pointing away from the electrode surface despite the negative electrode polarization. Finally, at more negative potentials, the decomposition reaction of the electrolyte is observed to yield a LiOH deposit at Au(111) surface.

Our work uncovers the detailed atomistic structure of the EDL in concentrated WiS electrolytes and identifies unique structural features at high negative electrode polarizations that differ from those in low-concentration electrolytes. These findings provide important insights for the molecular-level understanding of interfacial Li-ion behaviors in the high concentration condition, such as the de-solvation process of $Li^+$ and $Li^+$ layered accumulation, which would be beneficial for the electrode surface engineering in WiS electrolyte systems.

## Methods

### Materials

Lithium bis(trifluoromethane sulphonyl) imide (LiN(SO₂CF₃)₂, LiTFSI, purity of 99.95%), sodium citrate (99%), sodium silicate solution (reagent grade), and gold(III) chloride hydrate (~50% Au basis) were purchased from Sigma-Aldrich. The (3-aminopropyl)trimethoxysilane (APTMS, 97%) was purchased from Alfa Aesar. All the chemicals were used as received.

In electrochemical experiments, ~ 3 mL of electrolyte was used. An Ag and a Pt wire were used as the quasi-reference reference and counter electrode, respectively. Both the Ag and Pt wires are 0.5 mm in diameter, ~40 mm in length, and 99.999% in purity.

### Synthesis of Au shell-isolated nanoparticles (SHINs)

Au core nanospheres with a diameter of ~55 nm were synthesized according to reference[42]. In brief, 1.4 mL of 1 wt% sodium citrate solution was added to 200 mL of boiling 1 wt% HAuCl₄ solution (obtained by dissolving the gold(III) chloride hydrate into ultra-pure water), and the mixed solution was refluxed for 1 h. Au SHINs were synthesized according to ref. 29. Typically, 30 mL of a freshly prepared Au sol were mixed with 0.4 mL of 1 mM APTMS solution, then 3.2 mL of 0.54 wt% sodium silicate solution with a pH of about 10 was added to the mixed solution under stirring. The solution was heated to 95 °C in a water bath and stirred continuously for 45 min to accelerate the coating of a 2 nm silica shell on the Au core nanospheres. The prepared Au SHINs were twice washed with ultra-pure water and then dispersed in ultra-pure water for in situ electrochemical Raman experiments.

## Electrochemical Raman spectroscopic measurements

In situ Raman measurements were performed on a home-built Raman spectrometer with a backscattering configuration (see Supplementary Fig. 1c for the setup). Excitation at 633 nm from a HeNe continuous-wave laser (Thorlabs) was focused through a 50× objective (Mitutoyo), and the signal was collected via an electron-multiplying CCD camera (Andor). The laser power at the sample surface was ~10 mW. A halogen lamp (Thorlabs) was used for the white field imaging of the sample surface, which facilitates the data acquisition during spectro-electrochemical measurement. All the spectra were calibrated using a neon spectral calibration lamp (Newport) and a standard silicon (111) wafer. An Au(111) electrode was used as a working electrode throughout the experiment. Meanwhile, a homemade spectro-electrochemical cell with an Ag/AgCl reference electrode and a Pt counter electrode was used in the electrochemical Raman spectro-scopic experiments. To extract the signal of interfacial water, a spec-trum at the potential where the potential-dependent signal vanished was used as a reference to obtain the difference spectra of interfacial water[40]. Herein, the background was removed via Origin 2020 soft-ware. Because of the weak Raman intensity of interfacial water in highly concentrated electrolytes at a single-crystal surface, a Savitzky–Golay smoothing method was applied to the Raman spectra of interfacial water for a better signal-to-noise ratio. The ohmic drop in the spec-troelectrochemical cell was compensated for each Raman spectrum. All potentials in this work are referenced to the PZC. The electro-chemical Raman spectroscopic measurements were carried out at 25 °C without a climatic/environmental chamber.

It is worth noting that the SHINERS method was performed on an atomically flat Au(111) electrode surface, for which the enhancement of the Raman signal is much lower than that in the conventional surface-enhanced Raman spectroscopy (SERS) on a roughened Au electrode[43]. At the same time, the amount of water in the highly concentrated electrolyte is lower than that in the dilute electrolyte. Hence, the signal-to-noise ratio of the Raman spectra is lower as compared to the con-ventional SERS measurements in dilute electrolyte[43].

## Electrochemistry measurements

Cyclic voltammetry (CV) and electrochemical impedance spectro-scopy (EIS) measurements were performed in a glass single-chamber electrochemical cell (see Supplementary Fig. 19 for the photograph of the cell) on an electrochemical workstation (AUTOALB). Argon gas was bubbled into the electrolyte for 15 min to remove dissolved oxygen prior to each experiment. A slow argon flow was kept above the elec-trolyte during the whole experiment process. An Au(111) electrode was used as a working electrode. In the electrochemical polishing process, the Au(111) electrode was electrochemically oxidized in 0.5 M $H_2SO_4$ solution and then was immersed in 1 M HCl solution to reduce the gold oxide. Then, the electrode was washed using a great amount of ultra-pure water. These procedures were repeated three times. Finally, the electrode was annealed with an $H_2$ flame and cooled down in an argon atmosphere before each experiment. A platinum wire was used as the auxiliary electrode and a silver wire was used as a quasi-reference electrode. After each experiment, the potential was calibrated to Ag/AgCl electrode, which was used in in situ Raman measurements. The electrochemistry measurements were carried out at 25 °C without a climatic/environmental chamber.

The capacitance curves were obtained by EIS potential scan within the EDL region. The EISs were measured at the potentiostatic mode within the frequency range from 1 to $1 \times 10^5$ Hz and with 5 points per decade, considering the data availability and time consumption. The corresponding Nyquist and Bode plots of Au(111) electrode in 21 m LiTFSI aqueous electrolyte are shown in Supplementary Fig. 4c, d. The amplitude of the exciting signal was set to 10 mV relative to the root-mean-square voltage ($V_{RMS}$). The potential increment step was set to 50 mV in the potential scan and a 40 s quiet time was applied before each measurement to stabilize the EDL. We selected 15.85 Hz as the measuring frequency after analyzing the full frequency impedance response in the EDL. Impedance data were then converted to capaci-tance by the following formula ($C$ is the calculated capacitance, $f$ is the frequency applied, and $Z''$ is the imaginary part of impedance):

$$C = \frac{1}{2\pi f^*(-Z'')} \qquad (1)$$

Currently, there are several methods to determine the PZC, i.e., measuring differential capacitance curve[44,45] and the laser-induced current transient techniques[46,47]. But the determination of PZC in WiS electrolytes is challenging and there is no explicit theory to predict where it is. In this work, the PZC was measured by EIS, which has been widely used since the 1980s[44,45]. The differential capacitance curve was plotted by measuring the capacitance at certain potentials within the electrical double-layer region. Based on the literatures[28,44,45,48,49], we assumed that the capacitive minimum at around −0.1 V vs. Ag/AgCl corresponds to the PZC.

## Molecular dynamics simulation

Molecular dynamics (MD) simulations were utilized to investigate the structure of water-in-salt electrolytes (7 m and 21 m LiTFSI) at Au(111) electrode surfaces, as shown in Fig. 1b. Specifically, the SPC/E model was used for water molecules[50]; and the LiTFSI was modeled with all-atom force fields, which has been shown to reproduce the experi-mentally measured properties[51]. The Au(111) single-crystal electrode was modeled with the force field from Halicioglu and Pound[52,53]. Since in the potential region for exploring the reverse of Raman frequency shift of interfacial water, there are no experimental results indicating the formation of SEI on the electrode surface unless the potential becomes very negative (< ~ −1.55 V vs. PZC), MD simulations adopted the Au(111) electrode surface, which has the same structure with the single-crystal Au(111) electrode we used in SHINERS and electro-chemical experiments. The simulation system contained 648 LiTFSI, 5152 water for 7 m, and 992 LiTFSI, 2632 water for 21 m, respectively. The size of the simulation system was chosen as long enough to reproduce the bulk-like state of the mixture in the central region between the two electrode surfaces.

All simulations were performed in the NVT ensemble with the MD package GROMACS[54]. The temperature was controlled through the Nosé-Hoover thermostat[55,56] at 300 K with coupling constants of 1 ps. The van der Waals term was calculated via direct summation with a cutoff distance of 1.2 nm; meanwhile, a similar cutoff length was adopted in the calculation of electrostatic interactions in real space. The long-range electrostatic interactions were computed via the PME method[57], with an FFT grid spacing of 0.1 nm and cubic interpolation for the electrostatic interaction in the reciprocal space. The equation of motion was solved with a leapfrog integration algorithm, with a time step of 2 fs. Specifically, the constant-potential method (CPM)[58–60] was employed to allow the fluctuations of the charges on electrode atoms to ensure an adequate description of the electrode polarization effects in the presence of electrolytes. For each simulation, the MD system was first heated at 500 K for 3 ns and then annealed to 300 K over a period of 2 ns, followed by another 10 ns to reach equilibrium. After that, a 10 ns production was performed for analysis. Each case was repeated three times with different initial configurations to certify the accuracy of the simulation results.

## Electric field calculation

The electric field experienced by interfacial water is determined based on MD-obtained trajectories. Instead of the inaccurate cutoff method, the algorithm adopted herein is upgraded by performing the PME method[57], where the long-range electrostatic interactions induced by the electrode, water, and ions have been calculated precisely. Such

analysis has been used in prior simulation work[53]. Technologically, the van der Waals interactions of all particles are turned off first, and then the total electric field ($E_{total}$) experienced by interfacial water is determined based on the rerun of MD-obtained trajectories. Then it is decomposed into contributions from the electrode ($E_{H_2O-electrode}$), water molecules ($E_{H_2O-H_2O}$), and LiTFSI ($E_{H_2O-LiTFSI}$). To calculate the electric field from water molecules, only the electrostatic interaction of water with water is retained with the van der Waals interaction of all particles and the electrostatic interaction from electrode and LiTFSI being turned off; thus, the electric field ($E_1$) experienced by interfacial water is obtained from the rerun of MD-obtained trajectory, which is referred as $E_{H_2O-H_2O}$ ($E_{H_2O-H_2O} = E_1$). As for the contribution from LiTFSI, the van der Waals interaction of all particles and the electrostatic interaction from the electrode are turned off, thus the electric field ($E_2$) experienced by interfacial water is obtained from the rerun of the MD-obtained trajectory. Therefore, the electric field induced by ions is obtained as: $E_{H_2O-LiTFSI} = E_2 - E_{H_2O-H_2O}$. Consequently, the electric field from the electrode is calculated as: $E_{H_2O-electrode} = E_{total} - E_{H_2O-H_2O} - E_{H_2O-LiTFSI}$. It is worth noting that the MD-obtained electric field can be used to interpret the experiment qualitatively.

### Reporting summary

Further information on research design is available in the Nature Research Reporting Summary linked to this article.

## Data availability

The data that support the findings of this study are available from the corresponding authors upon reasonable request.

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

## Acknowledgements

The authors thank J.D.Wu for helpful discussions. This work was supported by the U.S. Air Force Office of Scientific Research (USAFOSR) under Grant No. FA9550-18-1-0420. M.C. and G.F. are supported by the National Natural Science Foundation of China (52106090 and 52161135104) and Hubei Provincial Natural Science Foundation of China (2020CFA093); G.F. is also supported by the Program for HUST Academic Frontier Youth Team.

## Author contributions

T.L., G.F., C.-Y.L., and M.C. conceived and designed the project, analyzed the results, and drafted the manuscript. C.-Y.L., J.M., and T.L. carried out the electrochemical Raman spectroscopy experiments and analysis. M.C. and G.F. performed the MD simulations. X.L. and H.D.A. prepared and characterized the single-crystal electrode. S.L. and J.Y. conducted the PZC measurement. All authors contributed to the final manuscript.

## Competing interests

The authors declare no competing interests.
