## [Peer Review File · Nature Communications]

REVIEWER COMMENTS

Reviewer #1 (Remarks to the Author):

A coordinated surface enhanced Raman and MD simulation study reported the changes in EDL of the recently discovered water-in-salt electrolytes (WiSE) from +0.9 to -1.5 V. Decomposition of the Raman signal is used to identify 0,1,2 donor O-H groups of water that are attributed to bulk-like water and Li(H₂O) with the reduced number of hydrogen bonds. Trends in Raman spectra for water on SiO₂ covered Au for -1.6 – 0 V region qualitatively agree with MD simulations of EDL on gold not covered by SiO₂. The influence of SiO₂ on the interfacial structure was not discussed. The unusual ‘H-up’ structure of interfacial water and the H-bonding network with TFSI⁻ anions at very negative potentials is reported as a major new finding but its importance is not demonstrated.

This is an interesting and informative study but three issues that prevent me from recommending this manuscript for publication

1) SEI formation is known to -0.7 V vs. SHE for this electrolyte and could be the reason behind the inversion of the Raman shifts below this potential unlike the effect found in MD simulations and attributed to the changes in Raman. This will be a completely different explanation of Raman data.

2) While the main conclusion is “Our work uncovers the detailed atomistic structure of the EDL in concentrated WiS electrolytes and identifies unique structural features at high negative electrode polarizations that differ significantly from those in low concentration electrolytes. These findings provide important insights for engineering the electrode surface to further enhance the performance of highly concentrated aqueous electrolytes for next-generation electrochemical energy storage and conversion devices.” Please elaborate how this important insight could be use “to further enhance the performance of highly concentrated aqueous electrolytes”. This will help to explain impact of the current study that is not quite clear from the current version of the manuscript.

3) Please demonstrate using CV as low scan rates and micro A/cm² currents that electrolyte is indeed stable at the surface used in experiments. Ref. 8 seem to show an onset of the reduction process at 2.6 V that would indicate that electrolyte is not stable for about half of the region shown in Figure 2 that Raman spectra should be interpreted with SEI growth and changes in chemical composition in mind instead of pore EDL changes.

Detailed comments:

Dubouis N, et al. (2018) The role of the hydrogen evolution reaction in the solid–electrolyte interphase formation mechanism for “Water-in-Salt” electrolytes. *Energ. & Env. Sci.* 11, pp 3491 reported the

growth of a crystalline compound, out of a large background signal, characterized by a peak at a diffraction angle of around 2θ is observed starting from -0.7 V vs. SHE. This is an onset of SEI formation that decomposes electrolyte that minimizes the effect of the charged interface on interfacial water. Thus, the changes of water structure at potentials below this potential is not just due to changes in the double layer but due to insulation of EDL by SEI and generation of additional species in the double layer.

Please magnify the reduction part of CV by x100 to clearly show an onset of reduction at low scan rates.

Could you repeat experiments at -0.7 V vs. SHE and hold for 24 hours with a limited electrolyte amount to confirm that LiOH and SEI do not form at this potential. The experimentally observed reduction in a blue shift at potentials below -1.2 V is likely due to growth of SEI that start insulating water molecules from the surface making them look more like bulk.

Note that addition of LiOH or NaOH to water also leads to red shifts of the water peaks. A control experiment showing the residual (difference) for addition of LiOH to WiSE as a function of LiOH salt concentration would be helpful in demonstrating that the observed trend is not due to degeneration of LiOH.

One of key conclusions is largely unsupported “The unusual ‘H-up’ structure of interfacial water and the H-bonding network with TFSI⁻ anions at very negative potentials may be used to tune the onset potential of water reduction reaction and to design a practical approach to protecting the aqueous electrolyte from performance degradation in high-voltage electrochemical energy devices, such as batteries and supercapacitors”. If such H-up structure would indeed prevent hydrogen evolution at -0.7 V vs. SHE, SEI generation at this potential would not have been observed.

MD simulations and experiments use different surfaces, please discuss ramifications.

Reviewer #2 (Remarks to the Author):

Review of “Unconventional interfacial water structure of highly concentrated aqueous electrolytes at negative electrode polarizations” by Li et al.

Comments on significance and overall quality

This is a highly interesting, well-constructed and largely well-interpreted contribution. The combination of molecular dynamics simulations with experimental surface-sensitive vibrational spectroscopy (SHINERS) is shown to be very powerful and has allowed the extraction of new insight into the electrode/solution interfacial region for water-in-salt (WiS) systems. The MD data is remarkable in its ability to define, in exquisite detail, the nature of all electrolyte components (cation, anion and water) at the interface. The MD simulations are qualitatively supported by the experimental Raman (SHINERS) data. The extracted information explains empirical observations of the large potential window of WiS systems, which is of tremendous practical importance, and verifies existing conjecture about the suppression of water electrolysis at negative electrode polarizations. In principle, the paper is well-suited for Nature Communications and I am confident that the paper will eventually warrant publication as a high-impact paper. However, there are places where I think some additional insight or explanation will greatly benefit the paper (see below).

Comments on Methodology/Experimental Details

I cannot comment on the MD (not my field of expertise) but the description of the spectroelectrochemistry is sound and provides sufficient information to be reproduced.

Comments on Data Analysis/Interpretation

SHINERS Spectra. Are the authors not surprised by the poor signal-to-noise of the OH stretching bands in the SHINERS data? I understand that the Raman cross-section of water is low, however, in comparison to the spectra of the TFSI⁻ anion shown in Supplementary Figure 10, the S/N of the water bands is remarkably poor. Using the data in Figure 3a, one can estimate that the surface density of water in the layer closest to the interface is $6-7 \times 10^{-10}$ moles cm^{-2} (NB – I did this by roughly determining the area of the peak at $z \sim 0.25$ nm in Figure 3a). This equates to a full monolayer of water so it is somewhat surprising to me that the OH stretching bands are so weak compared to SERS spectra of water adsorbed on metal electrodes (see for example Phys. Chem. Chem. Phys., 2010,12, 2493-250).

I assume that the weaker H-O-H bending mode is completely unobservable due to low S/N? The authors should mention this explicitly in the paper and comment on the overall S/N of the water signals.

Determination of the potential of zero charge. The authors use the minimum in the differential capacity curves they measure to assign the pzc to ~ -0.1 V vs Ag/AgCl. However, the authors rightly cite the works of Kornyshev and Oldham and indicate that a capacitive minimum is not predicted for ionic liquids or

highly concentrated electrolytes (see also the work by Bazant cited in reference 4). Neither Gouy-Chapman-Stern theory nor Kornyshev's theory for ionic liquids predict a capacitive minimum for highly concentrated electrolytes. The authors seem to be aware of this yet have chosen to ignore this complexity in the current paper. I note that others (e.g. Zhang et al ACS Appl. Energy Mater. 2020, 3, 8, 8086–8094) have made similar claims about a capacitive minimum corresponding to the pzc in WiS systems but it does not seem to be at all justified. Accurate determination of the pzc is critical to link the experimental SHINERS data with the MD simulations in the current work. I understand that determining the true pzc experimentally is difficult (perhaps it could be done with the laser-induced current transient technique) and it may be beyond the scope of the present study. However, the authors should address this point in the main part of the text and describe how uncertainty in the true value of the pzc affects their interpretation of their data.

Molecular Insights into the Electric Field Change. Figure 5b shows a plot of the simulated relationship between the electrode charge density and the applied potential. The slope of this line gives a constant differential capacity of $\sim 7 \mu\text{F cm}^{-2}$. Where does this number come from? How are the authors able to link applied potential and surface charge density in the MD simulations? Why would the electronic surface charge be expected to show a monotonic dependence on applied potential (relative to the pzc) when the experimentally measured capacity is clearly not constant with potential? (cf Supplementary Figure 3 with the slope of Figure 5b). The determination of the surface charge density underpins the arguments on the various contributions to the total field acting on the inner most layer of water. The total electric field is used to link the simulated data to the experimental Raman data (cf Raman shifts in Figure 2b and simulated E fields in Figure 2d) and explain the Stark shifts observed. However, the MD simulation seems to assume inherently different physics as evidenced by the differences in simulated and experimental charge-potential-capacity relationships.

Minor comments

There are at least two small typos.

Page 15 – “indicative”

Page 25 – “filed”

Reviewer #3 (Remarks to the Author):

This work concerns the interfacial water structure of highly concentrated water-in-salt (WiS) electrolytes, specifically, the electrical double layer structure of 21-m LiTFSI WiS electrolytes on the

Au(111) electrode surface. The authors combined in situ shell-isolated nanoparticle-enhanced Raman spectroscopy, known as SHINERS, with classical molecular dynamics (MD) simulations to investigate the OH stretch bands of water at the electrolyte-Au interface. The hydrogen-bonding structure and dynamics of interfacial water on Li-ion battery electrodes are important properties that change with applied electric potential. However, the methods they used to investigate these phenomena may not be suitable because the perturbation induced by Au nanoparticles placed on the surface could be strong enough to affect the interfacial water structure and ion distribution on the Au surface with a nonzero potential. In addition, the interpretation of the SHINERS Raman spectra with three bands representing different water species is not convincing. Therefore, I cannot recommend its publication in Nat. Comm.

Major comments:

1) The thin SiO₂ layer of the core/shell Au/SiO₂ nanoparticle (NP) is used to insulate the Au particle, where the thickness of the SiO₂ layer is approximately 2 nm, and the diameter of the core Au nanoparticle is about 57 nm. The SERS effect on the water OH stretching vibrations is likely to be large when water molecules are between the core/shell Au NP and Au electrode surface. From the SERS OH spectra, the authors tried to investigate the electric double layer (EDL) on the Au surface. However, there could be many inherent complicating factors due to the experimental design itself. First, the confinement effects on water (H-bonding) structure, ion-ion interaction, and ion-water interactions could be considerable. Second, when an electric potential is applied, the induced polarization of Au NP – note that the size of Au NP is huge as compared to a single water molecule or ions in the SERS region – could be non-negligible and would strongly affect those intermolecular interactions. Therefore, even though its surface is somewhat insulated by the SiO₂ layer, the perturbation induced by the presence of Au NP on the water structure, dynamics, and water-ion interaction strengths cannot be ignored, suggesting that the Raman spectra of those water molecules under such a strong perturbation could be very different from that of interfacial water molecules without the presence of Au NP on the Au electrode surface. Perhaps, what the authors have studied is not a realistic EDL of electrolyte on Au surface but the electric field change in the nanometric gap between Au NP and Au surface upon changing electric potential, which affects intermolecular interactions and solution structures.

2) The authors used MD simulation results to interpret their SHINERS Raman spectra of confined water molecules between Au NP and Au electrode surface. However, their MD simulations were performed for electrolyte-Au electrode composite systems without the huge Au NP on top of the Au electrode surface. Their interpretation of experimental results heavily relied on such simulations. Since the two composite systems considered experimentally and computationally are hugely different, one should not make use of MD simulation results to interpret the experimentally measured SHINERS spectra quantitatively.

3) They fit the SHINERS spectra with considering three bands that are assigned to the OH stretch mode of water molecules with $N_{\text{donor}} = 0, 1, \text{ and } 2$. This assignment was again made after analyzing and comparing their MD simulation results. However, such band assignments are not consistent with the previous experimental observations where OH stretch frequencies were found to be dependent on its H-bonded partners, e.g., water and TFSI⁻. For example, the OH frequency of OH...water differs from OH...TFSI⁻, but they are assumed to be identical (Figure 2f(ii)). Furthermore, there is no experimental evidence indicating that three (not two nor four) bands should be used to analyze the Raman spectra quantitatively.

4) The authors made a statement (in Abstract and throughout the main text) that the entire SHINERS spectra can be fitted with three bands. They justified the validity of this fitting analysis by comparing their MD simulation results. Again, because of the issue (2) above, such Raman spectral analyses are questionable.

5) The main finding in this paper is the notable change of the OH peak frequency as the electric potential changes from 0 to -1.55 V (Fig. 2b). The E-field amplitude calculated from the MD simulations also shows the same trend. The authors jumped to the conclusion that the local solution structures in the vicinity of the Au surface obtained from the MD simulations faithfully reflect those in the region confined by Au NP and Au surface. But, again, their MD ignored the effect of induced polarization of Au NP on the spatially confined electrolyte solution.

Minor comments:

1) In Fig. S4, three Raman spectra are shown. What do the authors mean by “bulk phase”? Also, it is not clear why the baseline increases with decreasing frequency (Raman shift).

2) In the first paragraph on page 9, the text refers to Supplementary Figs. 7 and 8, the captions of which say ‘simulated accumulative number densities’. Here, the meaning of ‘accumulative’ is ambiguous. What is the difference between ‘accumulated number density’ and just ‘number density’? Also, are there any quantitative criteria to define the ‘interfacial region’?

3) In Figs. 3e and 3f, the authors assign the peak near $\theta_{\text{Dipole}} = 60$ degrees as ‘H-up’ configuration. However, a water molecule satisfying the condition $\theta_{\text{Dipole}} = 60$ can have another rotational degree of freedom with the rotation axis along the direction of the water dipole so that the water plane makes 60 degrees with respect to the surface normal, in which case the OH bonds are rather far from the up direction. A kind of two-dimensional plot employing both θ_{Normal} and θ_{Dipole} would be more helpful to see the importance of the ‘H-up’ configuration.

4) The last paragraph on page 14 with Fig. 4 gives the argument that under low polarization, the interfacial water molecules adopt parallel and H-down structures, and under high polarization, ~30% of the Li-bound water adopts ‘H-up’ configuration, and the remaining Li-bound water molecules are in ‘H-down’ structure. I think this is not fully consistent with the results shown in Fig. 3b. The ‘H-down’ structure population is quite small under low polarization. The most conspicuous change in Fig. 3b with increasing polarization is the increase of the peak in the first layer, and the change in the third layer is quite minor. Also, I note that the strongest peak under high polarization in Fig. 3b is still the second peak corresponding to the parallel OH bond. The plot in Fig. 3b shows that a significant number of parallel OH bonds rotates to give the ‘H-down’ OH bonds as the polarization increases, while a small number of parallel OH bonds rotates to give the ‘H-up’ OH bonds. A single water molecule has two OH bonds, and a single water molecule can have a parallel OH and an ‘H-up’ or ‘H-down’ OH bond. Therefore, the estimated percentages of parallel, H-up, and H-down OH bonds would be informative, along with the percentages of ‘H-up’ and ‘H-down’ water molecules. Also, the authors’ argument seems to imply that the Li-bound water molecules in a parallel structure are nearly non-existent under high polarization, which needs to be more explicitly stated if that is the observation from the MD simulation.

5) I think one missing information regarding the 'H-up' structure is what percentage of the 'H-up' OH bonds are involved in hydrogen bonding and what percentage of them are dangling OH bonds. In addition, the authors say on page 14 that ~30% of the Li-bound water molecules are in an 'H-up' structure under high polarization. But, a more significant quantity would be the difference in the percentage of the 'H-up' configuration between low and high polarization. So, the percentage of 'H-up' structure in low polarization also needs to be given for comparison, although the change of the 3rd peak in Figure 3b provides qualitative information.

Reviewer #1 (Remarks to the Author):

General Comment:

A coordinated surface enhanced Raman and MD simulation study reported the changes in EDL of the recently discovered water-in-salt electrolytes (WiSE) from +0.9 to -1.5 V. Decomposition of the Raman signal is used to identify 0,1,2 donor O-H groups of water that are attributed to bulk-like water and Li(H₂O) with the reduced number of hydrogen bonds. Trends in Raman spectra for water on SiO₂ covered Au for -1.6 – 0 V region qualitatively agree with MD simulations of EDL on gold not covered by SiO₂. The influence of SiO₂ on the interfacial structure was not discussed. The unusual ‘H-up’ structure of interfacial water and the H-bonding network with TFSI- anions at very negative potentials is reported as a major new finding but its importance is not demonstrated. This is an interesting and informative study but three issues that prevent me from recommending this manuscript for publication

General Reply:

We thank the Reviewer very much for evaluating our work as an interesting and informative study, and the Reviewer’s comments indeed help to significantly improve the quality of our work. The point-to-point replies are given below.

Comment 1:

1) SEI formation is known to -0.7 V vs. SHE for this electrolyte and could be the reason behind the inversion of the Raman shifts below this potential unlike the effect found in MD simulations and attributed to the changes in Raman. This will be a completely different explanation of Raman data.

Reply 1:

Thank the Reviewer very much for the insightful comment. In our work, the WiS/Au(111) interfaces are investigated, and the reverse frequency shift of the OH stretching mode of interfacial water was observed at ~ -1.1 V vs. PZC (*i.e.*, ~ -1.0 V vs. SHE). Meanwhile, it is worth noting that, as shared by this Reviewer, Dubouis N, *et al.* reported the formation of SEI (LiOH) at -0.7 V vs. SHE (Ref: *Energy Environ. Sci.*, 11, 3491-3499, 2018), using Pt electrode, which is a different system to our research work. Herein, to testify whether there is a formation of the solid-electrolyte interphase (SEI) within the potential region showing the inverse Raman shifts, we obtained the Raman spectra from Au electrodes, which were applied by potential of -0.8 V vs. PZC (*i.e.*, -0.7 V vs. SHE) for 24 hours and

-1.2 V vs. PZC for 30 min, respectively. As shown in Figs. R1a and b, only a weak Raman signal of the residual electrolyte (S-N-S bending mode of LiTFSI at $\sim 750\text{ cm}^{-1}$) is recorded on the electrode surface. After the rinse of the electrode using ultra-pure water (Figs. R1c and d), there is no observable Raman signal of adsorbed species on the electrode surface. Hence, there is no spectroscopic evidence that SEI (such as LiOH, LiF, and Li_2CO_3 , *etc.*) is formed before and after the potential showing the occurrence of the reverse frequency shift. In addition, in our *in situ* Raman spectroscopic study shown in Fig. 2a in the main text of manuscript, the formation of SEI is observed till the potential becomes more negative than $\sim -1.55\text{ V vs. PZC}$ as indicated by the appearance of LiOH at $3,665\text{ cm}^{-1}$.

Figure R.1. (a-d). *Ex situ* Raman spectra of electrode surface after holding the potentials at -0.8 V for 24 hours (blue) and -1.2 V for 30 min (black) in the low (a and c) and high (b and d) frequency regions, respectively. Raman spectrum of 21 m WiS electrolyte is presented in a for comparison. In a and b, the Raman spectra were obtained without rinse of electrode surface, while in c and d, the same electrode surface was rinsed with ultra-pure water and then dried in argon. The peak marked by asterisk in a is from the residual electrolyte (S-N-S bending mode of LiTFSI at $\sim 750\text{ cm}^{-1}$) on the surface.

Figure R2. (a and c) AFM images of Au(111) electrode surface after holding in 21 m LiTFSI electrolyte at ~ -1.2 V (a) and ~ -1.8 V (c) for 5 min, respectively. (b and d) The height profiles of the Au(111) electrode surface corresponding to the black lines in a (b) and c (d), respectively. The white arrow in a and the black arrow in b remark the Au monatomic step. The fluctuation of the background in (b) is due to the noise from the instrument.

To further verify the absence of SEI revealed by Raman spectroscopy, we carried out atomic force microscope (AFM) measurements to examine the surface morphology of Au(111) electrode after holding at about -1.2 and -1.8 V vs. PZC. As revealed in Figs. R2 a and b, the electrode surface is still atomically flat after holding at -1.2 V for 5 min, consistent with the observation from Raman spectroscopic measurements, since the characteristic monatomic steps of Au(111) surface (a height of ~ 0.3 nm) can be clearly seen by the AFM measurement, indicating the high surface smoothness (Ref: *J. Am. Chem. Soc.*, 132, 13657–13659, 2010). On the contrary, as shown in Figs. R2 c and d, the surface morphology of electrode after holding at -1.8 V became more roughened with several valley-like

regions (a dip of ~ 3.4 nm), strongly suggesting the formation of SEI, which again agrees with Raman spectroscopic measurements (Figure 2a).

Briefly, based on the Raman spectroscopic and AFM experiments, the formation of SEI is not the reason for the inversion of Raman peak of interfacial water in the potential region we studied.

Revision:

The Figs. R1 and R2 are added in the updated Supporting Information as **Supplementary Figs. 8 and 9**.

On **page 9**, the discussion is added:

“In addition, we also hold the potential at -0.8 and -1.2 V on Au electrodes for 24 hours and 30 minutes, respectively, and then carried out *ex situ* Raman spectroscopic measurements to probe the electrode surface. As shown in Supplementary Fig. 8, there is no observable Raman signal of SEI (*i.e.*, the deposition of LiOH) on the electrode surface at such potentials. Furthermore, the atomic force microscope (AFM) measurements were performed to study the surface morphology of electrode surface after holding at about -1.2 and -1.8 V. As revealed in Supplementary Figs. 9a and b, the electrode surface is still atomically flat after holding at ~ -1.2 V for 5 min, where the characteristic monatomic steps of Au(111) surface can be clearly seen by the AFM measurement, indicating the high surface smoothness. In sharp contrast, after holding at ~ -1.8 V, the surface morphology of the electrode became much roughened with several valley-like regions (Supplementary Fig. 9c and d), suggesting the formation of SEI at this potential.”

Comment 2:

2) While the main conclusion is “Our work uncovers the detailed atomistic structure of the EDL in concentrated WiS electrolytes and identifies unique structural features at high negative electrode polarizations that differ significantly from those in low concentration electrolytes. These findings provide important insights for engineering the electrode surface to further enhance the performance of highly concentrated aqueous electrolytes for next-generation electrochemical energy storage and conversion devices.” Please elaborate how this important insight could be use “to further enhance the

performance of highly concentrated aqueous electrolytes”. This will help to explain impact of the current study that is not quite clear from the current version of the manuscript.

Reply 2:

Thanks for the Reviewer’s comment. Our study provides a detailed atomistic structure of the electrical double layer and the hydrogen-bonding network of Li⁺-bound water in highly concentrated electrolyte, which differ from those in dilute electrolyte. This study could be used to uncover the interfacial Li-ion behaviors in the high concentration condition, such as the de-solvation process of Li⁺ and Li⁺ layered accumulation, which would be beneficial for engineering the electrochemical reaction in highly concentrated aqueous electrolyte. Accordingly, we have revised the related conclusion as following:

Revision:

On **page 21**, the main conclusion has been revised:

“These findings provide important insights for the molecular-level understanding of interfacial Li-ion behaviors in the high concentration condition, such as the de-solvation process of Li⁺ and Li⁺ layered accumulation, which would be beneficial for the electrode surface engineering in WiS electrolyte systems”.

Comment 3:

3) Please demonstrate using CV as low scan rates and micro A/cm² currents that electrolyte is indeed stable at the surface used in experiments. Ref. 8 seem to show an onset of the reduction process at 2.6 V that would indicate that electrolyte is not stable for about half of the region shown in Figure 2 that Raman spectra should be interpreted with SEI growth and changes in chemical composition in mind instead of pore EDL changes.

Reply 3:

Thanks for this valuable suggestion. In the cyclic voltammogram (CV) represented as the inset of Fig. 1a in the first-version manuscript (copied as Fig. R3a), the positive potential limit is up to ~2 V, which have already induced the oxidation of electrode surface and the decomposition of electrolyte, as indicated by the large anodic current flow. As a result, a reduction wave of AuO_x occurs at around +0.4

V (Ref: *J. Am. Chem. Soc.*, 137, 7648-7651, 2015), followed by another reduction peak. In order to avoid the interference due to the intermediates from the electro-oxidation reactions, we limited the potential region for our *in situ* Raman spectroscopic study, and a lower scan rate at 10 mV/s was used as suggested. In addition, a flow of argon was used throughout the experiment to remove any residual oxygen in the solution.

First, as shown in Fig. R3b, in the CV from $\sim +0.2$ to -0.7 V, the current value is only around $4 \mu\text{A}/\text{cm}^2$, which is due to the charging of electrical double layer at the Au/WiS interface (Ref: *J. Phys. Chem. C*, 117, 205-212, 2013). Importantly, it is clear that the large reduction peak at -0.2 V in Fig. R3a disappears. Therefore, it is reasonable to attribute the reduction peak at -0.2 V in the CV in Fig. R3a to the reduction of the intermediates from the electro-oxidation reactions in the positive potential sweep toward $\sim +2$ V and also the possible reduction of oxygen-containing species.

Figure R3. Cyclic voltammograms of Au(111) in 21 m LiTFSI electrolyte within the different potential ranges. In (a), the CV is the same as the one in Fig. 1a of the first-version manuscript, where the scan rate and the unit of current density are 50 mV/s and mA/cm^2 . In (b) and (c), the scan rate and the unit of current density are changed to 10 mV/s and $\mu\text{A}/\text{cm}^2$ as suggested by the Reviewer.

Next, we also performed the CV toward a more negative potential at -1.4 V. As shown in Fig. R3c, a cathodic current starts at ~ -0.8 V, which can be attributed to the hydrogen evolution reaction and also the reduction of TFSI (Ref: *Science* 350, 938-943, 2015). However, as revealed by the *in situ* and *ex situ* vibrational spectroscopic experiments in Fig. 2a of manuscript and Fig. R1, together with the AFM measurements shown in Fig. R2, the SEI formation starts to occur at a more negative potential of about -1.55 V vs. PZC (~ -1.65 V vs. Ag/AgCl).

Hence, the potential-dependent frequency shift of Raman signal is not due to the formation of SEI in this work. Instead, the formation of SEI can be monitored by the appearance of Raman signal of LiOH at $\sim 3,665\text{ cm}^{-1}$ at potential negative than $\sim 1.55\text{ V vs. PZC}$ (Fig. 2a in manuscript). The growth of SEI will cause the rapid decrease of Raman intensity as the surface is covered, which is further evidenced by disappearance of interfacial water signal at $\sim 1.8\text{ V vs. PZC}$.

Revision:

In Fig. 1a of the first-version manuscript, the positive potential limit at $\sim +2\text{ V}$ can induce further electro-oxidation of both electrode and electrolyte, causing an additional reduction in the CV. To avoid this interference, we removed the CV in Fig. 1a, and added the new CVs with the optimized potential regions (Figs. R3 b and c) in the revised Supporting information as **Supplementary Fig. 3**.

Supplementary Fig. 3 | Cyclic voltammogram of Au(111) in 21 m LiTFSI electrolyte within the different potential ranges. In both (a) and (b), the scan rate is 10 mV/s.

Comment 4:

4) Dubouis N, et al. (2018) The role of the hydrogen evolution reaction in the solid–electrolyte interphase formation mechanism for “Water-in-Salt” electrolytes. *Energ. & Env. Sci.* 11, pp 3491 reported the growth of a crystalline compound, out of a large background signal, characterized by a peak at a diffraction angle of around 20° is observed starting from -0.7 V vs. SHE . This is an onset of SEI formation that decomposes electrolyte that minimizes the effect of the charged interface on interfacial water. Thus, the changes of water structure at potentials below this potential is not just to due changes in the double layer but due to insulation of EDL by SEI and generation of additional species in the double layer.

Reply 4:

Thanks for the comment of the Reviewer. In the literature shared by the Reviewer, the peak at a diffraction angle of around 20° is due to the formation of LiOH at -0.7 V vs. SHE on a gas diffusion layer electrode, as shown in the Fig. R4 copied from the literature (Ref: *Energy Environ. Sci.* 11, 3491-3499, 2018, Fig. 5).

Figure R4 (a) *Operando* XRD experiments on a gas diffusion layer electrode in 20 m LiTFSI. (b) Comparisons of XRD patterns of a gas diffusion layer electrode held at 1.8 V vs. SHE for 10 min and rinsed (red) the LiOH reference sample (grey). Figure was copied from the literature as the Reviewer suggested (Ref: *Energy Environ. Sci.* 11, 3491-3499, (2018), Fig. 5). The diffraction angle of around 20° is due to the formation of LiOH on the electrode.

In our work, as shown by the *in situ* experiments in Fig. 2 of our manuscript, the transition of Raman shift of interfacial water occurs at ~ -1.1 V vs. PZC. However, the Raman signal of SEI (*i.e.*, the deposition of LiOH) is not observed until the potential is at ~ -1.55 V or more negative vs. PZC. Furthermore, we also hold the potential at -0.8 and -1.2 V vs. PZC (-0.7 and -1.1 V vs. SHE) for a period of time and then acquired the Raman spectra of the electrode surface. As shown by the Raman spectra in Fig. R1, there is no Raman feature of the formation of SEI (such as LiOH, LiF, and Li₂CO₃,

etc.) before and after the potential that the frequency reversion occurs. Meanwhile, the AFM measurements in Fig. R2 show there is no formation of film on the electrode surface after holding at -1.2 V. Herein, in our work, the transition of Raman frequency shift is not due to the formation of SEI. In particular, we also used the subtraction method to remove the bulk water. As a result, the spectra obtained are potential-dependent signals from the electrical double layer (Refs: *Nat. Mater.* 18, 697-701, 2019; *J. Chem. Soc., Faraday Trans.* 92, 3829-3838, 1996). When the surface is insulated by SEI, *i.e.*, negative than -1.83V, it would not be able to observe the potential-dependent signal after the subtraction, which is consistent with the disappearance of interfacial water signal at -1.83 V.

Comment 5:

5) Please magnify the reduction part of CV by x100 to clearly show an onset of reduction at low scan rates

Reply 5:

Thanks for the suggestion of the Reviewer. We have magnified the reduction part of CV by more than 100 times, as shown in Fig. R5 a and b. The CV within a narrow potential region from +0.2 to -0.7 V is presented in Fig. R5c for comparison. It should be noted that the reduction current density in the potential region from +0.2 to -0.7 V is less $\sim 7 \mu\text{A}/\text{cm}^2$, which is attributed to charging/discharging of the electrical double layer (Ref: *J. Phys. Chem. C*, 117, 205-212, 2013). In the more negative potential region, a reduction wave occurs at ~ -0.8 V, which is due to the reductions of H_2O and TFSI^- (Ref: *Science* 350, 938-943, 2015). However, as we replied to Comment #3 of this Reviewer, although the electrolyte is reduced starting at ~ -0.8 V *vs.* Ag/AgCl, the SEI is formed till the potential negative than ~ -1.55 V *vs.* PZC, as evidenced by both the *in situ* and *ex situ* vibrational spectroscopic experiments (Fig. 2a in manuscript and Fig. R1) and the AFM measurements (Fig. R2).

Figure R5. Cyclic voltammograms of Au(111) in 21 m LiTFSI electrolyte. The cathodic part of CV (remarked by the dotted rectangle in **a**) has been magnified by more than 100 times and it is presented in **(b)**. The CV within the electrical double layer region is represented in **(c)**. The scan rate is 10 mV/s.

Comment 6:

6) Could you repeat experiments at -0.7 V vs. SHE and hold for 24 hours with a limited electrolyte amount to confirm that LiOH and SEI do not form at this potential. The experimentally observed reduction in a blue shift at potentials below -1.2 V is likely due to growth of SEI that start insulating water molecules from the surface making them look more like bulk.

Reply 6:

Thanks for the useful suggestion of the Reviewer. As we replied in comments #1 and #4 of this Reviewer, we hold the potentials at -0.8 V vs. PZC (-0.7 V vs. SHE) for 24 h and -1.2 V vs. PZC for 30 min, and then probed the electrode surface by Raman spectroscopy. As shown in Fig. R1, there are no Raman features indicating the formation of SEI on the electrode surface. At the same time, the AFM measurements in Fig. R2 reveal that the surface is still atomically flat at -1.2 V, while the SEI was observed at the more negative potential at ~ -1.8 V (in the *in situ* Raman experiment in Fig. 2a in manuscript, the LiOH was observed at ~ -1.55 V).

In addition, we used the subtraction method to remove the bulk water. As a result, the spectra obtained are potential-dependent signals from the electrical double layer (Refs: *Nat. Mater.* 18, 697-701, 2019; *J. Chem. Soc., Faraday Trans.* 92, 3829-3838, 1996), it is not as commented by the Reviewer, "...start insulating water molecules from the surface making them look more like bulk."

Revision:

The Fig. R1 and Fig. R2 are added in the **updated Supporting Information** as **Supplementary Figs. 8 and 9**.

On **page 9**, the discussion is added.

“In addition, we also hold the potential at -0.8 and -1.2 V on Au electrodes for 24 hours and 30 minutes, respectively, and then carried out *ex situ* Raman spectroscopic measurements to probe the electrode surface. As shown in Supplementary Fig. 8, there is no observable Raman signal of SEI (*i.e.*, the deposition of LiOH) on the electrode surface at such potentials.”

Comment 7:

7) Note that addition of LiOH or NaOH to water also leads to red shifts of the water peaks. A control experiment showing the residual (difference) for addition of LiOH to WiSE as a function of LiOH salt concentration would be helpful in demonstrating that the observed trend is not due to degeneration of LiOH.

Reply 7:

Thanks to the Reviewer for the suggestion. To study the LiOH on the effect of Raman frequency shift, we have performed an *in situ* experiment by directly applying a bias at ~ -1.6 V vs. PZC onto a fresh Au electrode to generate the LiOH on an electrode surface. Instead of directly adding LiOH, we hold the potential of ~ -1.6 V vs. PZC for different amount of times (1 and 14 min) to generate different amount of LiOH on the surface. As shown in Figure R6, the LiOH film thickness increases from 1 min to 14 min, as indicated by the LiOH peak amplitude at $\sim 3,665$ cm^{-1} . However, the ν_{OH} of interfacial water signals (the broad peak at $\sim 3,300$ cm^{-1}) are insensitive to thickness of the LiOH layer.

As shown in Fig. 2a in manuscript, the LiOH thickness increases during the negative potential scan. Therefore, the observed Raman shift trend cannot be attributed to the decomposition of LiOH.

Figure R6. (a). *In situ* Raman spectra of interfacial water with the formation and growth of LiOH by immediately applying the potential at ~ -1.6 V on a fresh Au electrode for a period of time, *i.e.*, 1 min (black curve) and 14 min (blue curve). The LiOH peak at $\sim 3,665$ cm⁻¹ was marked by dotted line. (b). Corresponding integration intensity of Raman signal of LiOH.

Comment 8:

8) One of key conclusions is largely unsupported “The unusual ‘H-up’ structure of interfacial water and the H-bonding network with TFSI⁻ anions at very negative potentials may be used to tune the onset potential of water reduction reaction and to design a practical approach to protecting the aqueous electrolyte from performance degradation in high-voltage electrochemical energy devices, such as batteries and supercapacitors”. If such H-up structure would indeed prevent hydrogen evolution at -0.7 V vs. SHE, SEI generation at this potential would not have been observed.

Reply 8:

Thanks for the comment. In our work, a “H-up” structure of interfacial water was found at very negative potential. According to a recently published work of humid ionic liquids, a dipole-up water structure was theoretically predicted to reduce the probability of electrons transferred from electrode to water, leading to a more negative potential for the hydrogen evolution reaction (Ref: *Nat. Commun.* 11, 5809, 2020, Pages 31 -34 in the corresponding Supporting Information). Based on this literature, the “H-up” interfacial water structure may be used to tune the potential of the hydrogen evolution reaction. In the revised manuscript, we have added the reference and modified this statement.

Revision:

On **page 20**, the discussion is made as:

“Recently, a ‘H-up’ water structure was predicted to lower the potential of hydrogen evolution reaction on the negative electrode in humid ionic liquids¹⁷. Therefore, ...”. Accordingly, a literature (Ref: *Nat. Commun.* 11, 5809, 2020) was added as reference #17 here.

Comment 9:

9) MD simulations and experiments use different surfaces, please discuss ramifications.

Reply 9:

Thanks for the helpful comment of the Reviewer. As shown in Fig. R1, as the potentials holding at -0.8 V vs. PZC (-0.7 V vs. SHE) for 24 h and -1.2 V vs. PZC for 30 min, there is no Raman features indicating the formation of SEI on the electrode. More importantly, the AFM image of Au(111) electrode after holding at -1.2 V shows that the surface is still atomically flat (Fig. R2a). Hence, in the potential region we studied the reverse of Raman frequency shift of interfacial water, there is no formation of SEI on the electrode surface unless the potential becomes very negative ($< \sim -1.55$ V vs. PZC).

Therefore, MD simulations used the Au(111) electrode surface, which is the same single crystal Au(111) electrode we used in Raman spectroscopy, CV, AFM, and EIS experiments. We have added this clarification in the Method section.

Revision:

In the method section (**page 29**), the clarification of the electrode surface used in MD simulations, spectroelectrochemical and electrochemical experiments are added.

“Since in the potential region for exploring the reverse of Raman frequency shift of interfacial water, there is no experimental results indicating the formation of SEI on the electrode surface unless the potential becomes very negative ($< \sim -1.55$ V vs. PZC), MD simulations adopted the Au(111) electrode

surface, which has the same structure with the single crystal Au(111) electrode we used in SHINERS and electrochemical experiments.”

Reviewer #2 (Remarks to the Author):

General Comment:

This is a highly interesting, well-constructed and largely well-interpreted contribution. The combination of molecular dynamics simulations with experimental surface-sensitive vibrational spectroscopy (SHINERS) is shown to be very powerful and has allowed the extraction of new insight into the electrode/solution interfacial region for water-in-salt (WiS) systems. The MD data is remarkable in its ability to define, in exquisite detail, the nature of all electrolyte components (cation, anion and water) at the interface. The MD simulations are qualitatively supported by the experimental Raman (SHINERS) data. The extracted information explains empirical observations of the large potential window of WiS systems, which is of tremendous practical importance, and verifies existing conjecture about the suppression of water electrolysis at negative electrode polarizations. In principle, the paper is well-suited for Nature Communications and I am confident that the paper will eventually warrant publication as a high-impact paper. However, there are places where I think some additional insight or explanation will greatly benefit the paper (see below).

Comments on Methodology/Experimental Details

I cannot comment on the MD (not my field of expertise) but the description of the spectroelectrochemistry is sound and provides sufficient information to be reproduced.

General Reply:

We thank the Reviewer very much for the appreciation and the high assessment of our work. We also want to thank the Reviewer for the insightful comments, which help to improve our manuscript and the interpretation. The point-by-point responses are given below.

Comments on Data Analysis/Interpretation

Comment 1:

SHINERS Spectra. Are the authors not surprised by the poor signal-to-noise of the OH stretching bands in the SHINERS data ? I understand that the Raman cross-section of water is low, however, in comparison to the spectra of the TFSI⁻ anion shown in Supplementary Figure 10, the S/N of the water bands is remarkably poor. Using the data in Figure 3a, one can estimate that the surface density of water in the layer closest to the interface is $6-7 \times 10^{-10}$ moles cm^{-2} (NB – I did this by roughly determining the area of the peak at $z \sim 0.25$ nm in Figure 3a). This equates to a full monolayer of water so it is somewhat surprising to me that the OH stretching bands are so weak compared to SERS spectra of water adsorbed on metal electrodes (see for example Phys. Chem. Chem. Phys., 2010,12, 2493-

250).

I assume that the weaker H-O-H bending mode is completely unobservable due to low S/N ? The authors should mention this explicitly in the paper and comment on the overall S/N of the water signals.

Reply 1:

Thanks for the comment of the Reviewer. Actually, the low S/N is due to two main reasons: (1) the amount of water in super concentrated water-in-salt electrolyte is very low, only ~20% in volume to that in the dilute aqueous electrolyte; (2) we used the SHINERS method on an atomically flat Au(111) single crystal electrode surface, the surface enhancement is low as compared to conventional surface-enhanced Raman spectroscopy (SERS) on a roughened Au electrode, which was used as working electrode in the literature the Reviewer mentioned (Ref: *Phys. Chem. Chem. Phys.*, 12, 2493-2502, 2010).

However, because the conventional SERS requires a nanostructured or a roughened metal surface, it is unable to obtain the Raman signal of adsorbed water on the atomically flat electrode surface. Therefore, although the signal is low, the SHINERS method is currently one of the very few methods capable of studying the atomic structure of electric double layer on a single crystal electrode surface by vibrational spectroscopy.

We also agree with the Reviewer that weaker H-O-H bending mode is technically difficult to observe. Accordingly, we have revised the manuscript on the discussion of S/N ratio of water signals using SHINERS method (in Method part).

Revision:

The discussion of the S/N ratio of water signals is added in the **Method part (page 27)**.

“It is worth noting that the SHINERS method was performed on an atomically flat Au(111) electrode surface, for which the enhancement of Raman signal is much lower than that in the conventional surface-enhanced Raman spectroscopy (SERS) on a roughened Au electrode⁴⁴. At the same time, the amount of water in the highly concentrated electrolyte is lower than that in the dilute electrolyte.

Hence, the signal-to-noise ratio of the Raman spectra is lower as compared to the conventional SERS measurements in dilute electrolyte⁴⁴.”.

Accordingly, a literature (Ref: *Phys. Chem. Chem. Phys.*, 12, 2493-2502, 2010) was added as reference #44 here.

Comment 2:

Determination of the potential of zero charge. The authors use the minimum in the differential capacity curves they measure to assign the pzc to ~ -0.1 V vs Ag/AgCl. However, the authors rightly cite the works of Kornyshev and Oldham and indicate that a capacitive minimum is not predicted for ionic liquids or highly concentrated electrolytes (see also the work by Bazant cited in reference 4). Neither Gouy-Chapman-Stern theory nor Kornyshev’s theory for ionic liquids predict a capacitive minimum for highly concentrated electrolytes. The authors seem to be aware of this yet have chosen to ignore this complexity in the current paper. I note that others (*e.g.* Zhang et al *ACS Appl. Energy Mater.* 2020, 3, 8, 8086–8094) have made similar claims about a capacitive minimum corresponding to the pzc in WiS systems but it does not seem to be at all justified. Accurate determination of the pzc is critical to link the experimental SHINERS data with the MD simulations in the current work. I understand that determining the true pzc experimentally is difficult (perhaps it could be done with the laser-induced current transient technique) and it may be beyond the scope of the present study. However, the authors should address this point in the main part of the text and describe how uncertainty in the true value of the pzc affects their interpretation of their data.

Reply 2:

We totally agree with the Reviewer that accurate determination of the PZC by experiments is difficult. Measuring the differential capacitance curve is one of the most commonly used methods to determine the PZC. (Refs: *Electrochim. Acta* 25, 1527-1529, 1980; *J. Electrochem. Soc.* 144, 3392-3397, 1997; *ACS Appl. Energy Mater.* 3, 8086-8094, 2020). However, for the concentrated electrolytes (*i.e.*, the WiS electrolyte studied in this work), the determination of PZC becomes more complicated. Although Kornyshev and Oldham’s theory predicted different shapes of differential capacitance curves in ionic liquids (Refs: *J. Phys. Chem. B*, 111, 5545-5557, 2007; *Chem. Rev.*, 114, 2978-3036, 2014), it is still challenging to clearly describe the interfaces between the condensed ions and the electrode. For instance, Kornyshev also gave a camel-shape differential capacitance curve denoting that the PZC lies

at the capacitive minimum, but it shifts positively when the chemical composition of ionic liquid is asymmetric (Ref: *Electrochim. Acta*, 225, 190-197, 2017).

We want to thank the Reviewer for the suggestion of using the laser-induced current transient technique, which could avoid the influence mentioned above (Refs: *J. Phys. Chem. B*, 105, 10669-10673, 2001; *Electrochem. Commun.*, 62, 44-47, 2016). However, this technique requires additional special instruments. Therefore, it is indeed beyond the scope of the present study as the Reviewer recognized.

Based on the literatures (Refs: *J. Phys. Chem. B* 111, 5545-5557, 2007; *J. Electroanal. Chem.* 613, 131-138, 2008), we assumed that the capacitive minimum corresponds to the PZC, and the PZC is found as around -0.1 V vs. Ag/AgCl, which is similar to that obtained by MD simulation. Therefore, in our work, the PZC indicated by the capacitive minimum is reasonable, and we added the above-mentioned discussion in the manuscript as the Reviewer suggested.

Revision:

In the **Method section** of electrochemistry measurements (**Page 28**), the following discussion is added.

“Currently, there are several methods for determining the PZC, including measuring differential capacitance curve^{45,46} and the laser-induced current transient techniques^{47,48}. But the determination of PZC in WiS electrolytes is challenging and there is no explicit theory to predict where it is. In this work, the PZC was measured by EIS, which has been widely used since the 1980s^{45,46}. The differential capacitance curve was plotted by measuring the capacitance at certain potentials within the electrical double layer region. Based on the literatures^{28,45,46,49,50}, we assumed that the capacitive minimum at around -0.1 V vs. Ag/AgCl corresponds to the PZC.”

Accordingly, several literatures were added as references #45, #46, #49 and #50 here.

Comment 3:

Molecular Insights into the Electric Field Change. Figure 5b shows a plot of the simulated relationship between the electrode charge density and the applied potential. The slope of this line gives a constant differential capacity of $\sim 7 \mu\text{F cm}^{-2}$. Where does this number come from? How are the authors able to link applied potential and surface charge density in the MD simulations? Why would the electronic surface charge be expected to show a monotonic dependence on applied potential

(relative to the pzc) when the experimentally measured capacity is clearly not constant with potential ? (cf Supplementary Figure 3 with the slope of Figure 5b). The determination of the surface charge density underpins the arguments on the various contributions to the total field acting on the inner most layer of water. The total electric field is used to link the simulated data to the experimental Raman data (cf Raman shifts in Figure 2b and simulated E fields in Figure 2d) and explain the Stark shifts observed. However, the MD simulation seems to assume inherently different physics as evidenced by the differences in simulated and experimental charge-potential-capacity relationships.

Reply 3:

Thanks for this insightful comment. Firstly, we would like to describe how we obtain the surface charge density from MD simulation. In MD simulation, we employ the constant potential method (CPM) to mimic the polarization effects from the electrode in the presence of electrolytes. (Refs: *Nat. Mater.*, 11, 306-310, 2012; *ACS Nano*, 9, 5999-6017, 2015; *Nat. Energy*, 1, 16070, 2016) Specifically, the CPM assumes Gaussian charge distribution for electrode atoms and point charges for electrolyte atoms. The tunable charges on electrodes (\mathbf{q}) are extra degrees of freedom to keep the electrodes equipotential. The electrostatic energy of the system, U_{ele} , is a function of atom positions (\mathbf{X}) and electrode charge, as (Refs: *Nat. Mater.*, 11, 306-310, 2012; *Nat. Energy*, 1, 16070, 2016; *Nat. Commun.* 11, 5809, 2020; *Nat. Comput. Sci.* 1, 725-731, 2021):

$$U_{ele}(\mathbf{X}, \mathbf{q}) = \frac{\mathbf{q}^T \mathbf{A} \mathbf{q}}{2} - \mathbf{q}^T \mathbf{B}(\mathbf{X}) + C(\mathbf{X})$$

The first and second terms in the right-hand side of the above equation represent the electrode-electrode and electrode-electrolyte interactions, respectively. The third term describes the electrolyte-electrolyte interaction. The Matrix A depends on the position of electrode atoms. Therefore, the relationship between the electrode charge density and the applied potential can be obtained through running MD simulations.

Although MD simulations have become a powerful technique to investigate the electrode/electrolyte interfaces by monitoring the electrosorption of electrolyte ions on the surfaces, we must admit that the MD simulations failed to quantitatively reproduce the capacitance on the metal surface, which is always smaller than the experiment (*J. Phys. Chem. B.*, 101, 8550-8558, 1997; *J. Phys. Chem. Lett.*, 9, 1927-1930, 2018; *Faraday Discuss.*, 141, 423-441, 2009; *ChemElectroChem*, 4, 216-220, 2017). It has been

experimentally measured that the capacitance of EDLs in aqueous electrolytes at metal electrodes is in the range of 20~50 $\mu\text{F}/\text{cm}^2$. (Ref: *J. Phys. Chem. Lett.*, 9, 1927-1930, 2018) For instance, a differential capacitance around 30 $\mu\text{F}/\text{cm}^2$ for aqueous electrolytes on the metal electrode was detected (Refs: *J. Phys. Chem. B.*, 101, 8550-8558, 1997; *J. Phys. Chem. Lett.*, 9, 1927-1930, 2018), in the same order of magnitude of our experiment (Supplementary Fig. 4). So far, simulation studies generally reported capacitances in the range of 5~10 $\mu\text{F}/\text{cm}^2$, such as the aqueous electrolytes or the ionic liquids on the metal electrodes (Refs: *Faraday Discuss.*, 141, 423-441, 2009; *ChemElectroChem*, 4, 216-220, 2017). This discrepancy could be interpreted as the limitation of MD simulation where conventional force fields may not accurately capture the chemisorption effects (Ref: *Faraday Discuss.*, 141, 423-441, 2009). Despite this limitation, MD simulations have qualitatively provided significant microscopic insights into the interpretation of electrochemistry experiments (Ref: *Nat. Mater.*, 19, 1096–1101, 2020). For instance, an oscillating ion distribution of the superconcentrated electrolyte was revealed near electrode surface by MD simulation, in qualitative agreement with the atomic force measurement results. (Ref: *Nat. Mater.*, 19, 1096–1101, 2020) A qualitative description has also been achieved to link the local structure and electric field with the vibrational modes of $\text{C}\equiv\text{N}$ on a nanostructured Au electrode utilizing MD simulation. (Ref: *J. Phys. Chem. C.*, 121, 22274-22285, 2017)

In terms of our work, although the magnitude of experimentally measured surface charge density is higher than the MD simulation, the surface charge density obtained from experiment does exhibit a monotonic dependence on applied potential, in line with the MD (Fig. R7). In this regard, it is reasonable to link the simulated data to the experimental data qualitatively (cf Raman shifts in Fig. 2b and simulated E fields in Fig. 2d in the manuscript).

Figure R7. Surface charge density (σ) as a function of potential for experiment (a) and MD simulation (b).

Revision:

The above discussion is added in the revised manuscript as:

“Gray line in (b) is to guide the eyes” (Caption of Fig. 5 in the maintext, Page 19);

“It is worth noting that the MD-obtained electric field is used to interpret the experiment qualitatively.”

Method part (Electric field calculation, page 31).

Comment 4:

Minor comments

There are at least two small typos.

Page 15 – “indicative”

Page 25 – “filed”

Reply 4:

Thanks for the comments, we have revised the manuscript accordingly.

Reviewer #3 (Remarks to the Author):

This work concerns the interfacial water structure of highly concentrated water-in-salt (WiS) electrolytes, specifically, the electrical double layer structure of 21-m LiTFSI WiS electrolytes on the Au(111) electrode surface. The authors combined in situ shell-isolated nanoparticle-enhanced Raman spectroscopy, known as SHINERS, with classical molecular dynamics (MD) simulations to investigate the OH stretch bands of water at the electrolyte-Au interface. The hydrogen-bonding structure and dynamics of interfacial water on Li-ion battery electrodes are important properties that change with applied electric potential. However, the methods they used to investigate these phenomena may not be suitable because the perturbation induced by Au nanoparticles placed on the surface could be strong enough to affect the interfacial water structure and ion distribution on the Au surface with a nonzero potential. In addition, the interpretation of the SHINERS Raman spectra with three bands representing different water species is not convincing. Therefore, I cannot recommend its publication in Nat. Comm.

General Reply:

We thank the Reviewer very much for her/his time on evaluating our work, and the reviewer's comments on Method help to solidify our work. The point-to-point replies are given below.

Comment 1:

The thin SiO₂ layer of the core/shell Au/SiO₂ nanoparticle (NP) is used to insulate the Au particle, where the thickness of the SiO₂ layer is approximately 2 nm, and the diameter of the core Au nanoparticle is about 57 nm. The SERS effect on the water OH stretching vibrations is likely to be large when water molecules are between the core/shell Au NP and Au electrode surface. From the SERS OH spectra, the authors tried to investigate the electric double layer (EDL) on the Au surface. However, there could be many inherent complicating factors due to the experimental design itself. **First**, the confinement effects on water (H-bonding) structure, ion-ion interaction, and ion-water interactions could be considerable. **Second**, when an electric potential is applied, the induced polarization of Au NP – note that the size of Au NP is huge as compared to a single water molecule or ions in the SERS region – could be non-negligible and would strongly affect those intermolecular interactions. Therefore, even though its surface is somewhat insulated by the SiO₂ layer, the perturbation induced by the presence of Au NP on the water structure, dynamics, and water-ion interaction strengths cannot be ignored, suggesting that the Raman spectra of those water molecules under such a strong perturbation could be very different from that of interfacial water molecules without the presence of Au NP on the Au electrode surface. Perhaps, what the authors have studied is not a realistic EDL of electrolyte on Au surface but the electric field change in the nanometric gap between Au NP and Au surface upon changing electric potential, which affects intermolecular

interactions and solution structures.

Reply 1:

Thanks for the Reviewer's insightful comment upon the SHINERS method. As is well known, it is intrinsically difficult to obtain Raman signal of interfacial water on an atomically flat electrode surface without plasmonic enhancement. In order to obtain the Raman signal from single electrode surface, the shell-isolated nanoparticles (SHINs) are spread onto the electrode surface for Raman enhancement. As shown by the schematic and scanning electron microscopy (SEM) image in Fig. R8, a sub-monolayer of SHINs can contact the Au(111) electrode surface in the SHINERS method.

Figure R8. Schematic of SHINERS method and the SEM image of Au(111) electrode surface uniformly covered by a sub-monolayer of SHINs.

Most importantly, in SHINERS method, the ultrathin silica shell is electrically inert, which can prevent electron transfer between the Au nanoparticle core and the electrode surface during electrochemical process (Refs: *Nature*, 464, 392–395, 2010; *Nat. Commun.* 7, 12440, 2016; *Nature*, 600, 81–85, 2021). As an example shown by the CV in Fig. R9a copied from Ref. (*Nature*, 464, 392–395, 2010), without the coating of SiO₂ shell, the Au nanoparticle core can be oxidized and then reduced as revealed by the characteristic oxidation at >1.2 V and reduction peak at ~ 0.9 V (red curve in Fig. R9a). On the contrary, in the CV of the electrode with SHINs (black curve in Fig. R9a), there are no oxidation and reduction peaks of Au, which indicates the electrically inert property of SHINs.

Some work has reported that the SHINERS method is suitable to study the property of electrical double

layer at the metal electrode. For instance, as indicated by the Fig. R9b adapted from literature (Ref: *Nat. Commun.*, 7, 12440, 2016), the CV of Pt(111) electrode covered by SHINs shows the similar features with that of a bare Pt(111) electrode in 0.1 mol/L HClO₄. In particular, the voltammogram of hydrogen adsorption/desorption (potential from 0.08 V to 0.5 V) and the phase transition of OH (potential from 0.5 V to 0.9 V) on Pt(111) electrode covered by SHINs are similar with that on bare Pt(111) electrode. All of the CVs of SHINs covered glassy carbon electrode (black curve in Fig. R9a) and Pt(111) electrode (red and blue curve in Fig. R9b) show the very small charging/discharging currents of the double layer capacitance, which also indicates the inert surface chemical property of the thin SiO₂ shell of SHINs.

Figure R9. (a) Cyclic voltammograms (CVs) of 55 nm Au nanoparticles (red line) and 55 nm Au@SiO₂ SHINs (black line) on glassy carbon (GC) electrodes in 0.5 M H₂SO₄ solution. Figure adapted from Ref: *Nature*, 464, 392–395, 2010. (b) CVs of Pt(111), of Pt(111) covered by Au@SiO₂ and of a Pt(111) electrode covered by Au@SiO₂ (SHINs) after treatment by the hydrogen evolution reaction, as used in SHINERS measurements in 0.1 mol/L HClO₄. Figure adapted from Ref: *Nat Commun* 7, 12440, 2016.

Therefore, literature suggests that the SHINERS method are suitable for the investigations of electrochemical interfaces, such as specific adsorption of SO₄²⁻, pyridine, and hydrogen (Refs: *Nature*, 464, 392–395, 2010; *J. Am. Chem. Soc.*, 137, 2400-2408, 2015; *J. Am. Chem. Soc.*, 142, 9439-9446, 2020), and electrocatalytic reactions (Refs: *Nat Commun*, 7, 12440, 2016; *J. Am. Chem. Soc.*, 141, 12192-12196, 2019; *J. Am. Chem. Soc.*, 142, 9735-9743, 2020), and more importantly and relevantly, the **structure of interfacial water and electrical double layer** (Refs: *Nat. Mater.* 18, 697-701, 2019;

Nature, 600, 81–85, 2021).

Figure R10. (a) The schematics of SHINERS method using different SHINs coverage by dropping different volume of as-prepared SHINs solution onto the electrode surface. The surface coverages of SHINs are calculated to be $\sim 11\%$ and $\sim 28\%$ for low and high coverage. (b) and (c). Raman spectra of adsorbed TFSI⁻ (b) and interfacial water (c) at Au(111) surface at -0.55 V vs. PZC with the low (black) and high (red) SHINs coverages.

To further explore the feasibility of SHINERS method on the EDL at the Au/WiS electrolyte interface, we compared the spectroscopic features obtained with **low and high** SHINs coverages on the Au(111) electrode surface.

As shown in the schematics in Fig. R10a, the coverage of SHINs can be simply tuned by the volume of as-prepared SHINs solution used in experiment. Herein, 10 and 25 μL of SHINs solutions were dropped onto the Au(111) electrode surface and dry in vacuum to obtain a low ($\sim 11\%$) and high ($\sim 28\%$) SHINs coverage on the electrode. If the SHINs have an influence on the water (H-bonding) structure, ion-ion interaction, and ion-water interactions, the Raman spectra of anions and interfacial water will change with the increase of SHINs. As revealed by the Raman spectra using different SHINs coverages at -0.55 V vs. PZC (Figs. R10b and c), the frequency and shapes of Raman peaks of both adsorbed TFSI⁻ anion and interfacial water are not influenced by the SHINs coverage. The main difference is that the intensity of spectra obtained with low SHINs coverage is weaker than those in the high SHINs

coverage case, which can be explained by the weaker surface enhancements provided by the smaller number of SHINs.

It is important to emphasize that we use the SHINER method to measure molecules within the EDL. This is done by using a subtraction method, in which the potential independent contribution of molecules outside of EDL have been subtracted out using the spectrum at +0.9 V vs. PZC as a reference (the spectrum at this potential shows negligible potential dependence). As a result, potential-dependent Raman signal of interfacial water can be revealed by the spectra after subtraction method. While we agree with the reviewer that the Au particle may affect the structure of electrolyte between the SHINparticle and the Au electrode, especially electrolytes near the SHIN particle, it is possible that their effect on the EDL of the Au electrode may be negligible. This notion appears to be supported by recent studies of EDL structure of metal electrodes (Refs: *Nature*, 600, 81–85, 2021; *Nat. Mater.*, 18, 697-701, 2019; *Frontiers of surface-enhanced Raman scattering* (eds Y. Ozaki, K. Kneipp, & R. Aroca), 163-192, 2014).

Therefore, the SHINERS method is suitable to study the electrical double layer and the solution structures under bias conditions.

Revision:

On **page 4**, the feasibility of SHINERS method for the electrochemical interface is added as:

“..., which is suitable for the investigations at electrochemical interface, such as specific adsorption of sulfate ion³⁰, pyridine³¹, and hydrogen²⁹, and most importantly and relevantly, enables an *in situ* molecular level probe of the structures of electrical double layer and interfacial water on single crystal electrode surfaces^{22,29,32}. ”

Accordingly, several literatures were added as references #29, #30, #31 and #32 here.

Comment 2:

The authors used MD simulation results to interpret their SHINERS Raman spectra of confined water molecules between Au NP and Au electrode surface. However, their MD simulations were performed for electrolyte-Au electrode composite systems without the huge Au NP on top of the Au electrode

surface. Their interpretation of experimental results heavily relied on such simulations. Since the two composite systems considered experimentally and computationally are hugely different, one should not make use of MD simulation results to interpret the experimentally measured SHINERS spectra quantitatively.

Reply 2:

Thanks for the Reviewer's comment. As described in response to previous comment, firstly, we would like to clarify that the Au nanoparticle core is electrostatically isolated by an electrochemically inert silica shell in the SHINERS (Refs: *Nature*, 464, 392–395, 2010; *Nat. Commun.* 7, 12440, 2016; *Nature*, 600, 81–85, 2021). Secondly, the Raman signal of water we presented in manuscript is mainly from the interfacial region, and negligibly from “confined water molecules”. Importantly, as discussed in the reply to Comment #1 from this Reviewer, the Raman spectroscopic experiments with different SHINs coverages in Fig. R10 reveal that the perturbation effect caused by SHINs is negligible in our study. Therefore, the interface studied by SHINERS method is the EDL in WiS electrolytes at the Au(111) electrode. Therefore, the SHINERS Raman spectra can be interpreted using MD simulations of interfaces between WiS electrolytes and Au(111) electrode in our work.

3) They fit the SHINERS spectra with considering three bands that are assigned to the OH stretch mode of water molecules with $N_{\text{donor}} = 0, 1, \text{ and } 2$. This assignment was again made after analyzing and comparing their MD simulation results. However, such band assignments are not consistent with the previous experimental observations where OH stretch frequencies were found to be dependent on its H-bonded partners, e.g., water and TFSI-. For example, the OH frequency of OH...water differs from OH...TFSI-, but they are assumed to be identical (Figure 2f(ii)). Furthermore, there is no experimental evidence indicating that three (not two nor four) bands should be used to analyze the Raman spectra quantitatively.

Reply 3:

We appreciate the Reviewer for this comment. In low-concentrated aqueous electrolytes, the Raman spectra of interfacial water have been assigned to three distinct components, describing three different types of OH stretching vibrations. (Refs: *Nat. Mater.* 18, 697–701, 2019; *Nature*, 600, 81–85, 2021). However, the Raman spectrum analysis of interfacial water at the electrode-WiS electrolyte interface has not been studied yet. Therefore, in our work, inspired by the previous work on the aqueous electrolyte in low concentration, the Raman spectroscopic analysis of interfacial water at the electrode-

WiS electrolyte interface was also assigned into three bands, considering the only change is the ratio of water to ions. Previous researches have demonstrated that the donor number of water molecules has a significant effect on the OH stretching frequency, with the frequency of OH stretching mode increasing as the donor number decreases. (Refs: *J. Am. Chem. Soc.*, 127, 15916-15922, 2005; *Nat. Mater.*, 18, 697–701, 2019; *Nature*, 600, 81–85, 2021; *Advances in infrared and Raman spectroscopy* Vol. 5 (eds Robin Jon Hawes Clark & Ronald E. Hester) Ch. 3, (Heyden, 1978)). In order to assist the assignments of the observed interfacial OH stretching spectra, MD simulations were performed to explore the structure of water molecules and the corresponding H-bond network at the electrode-WiS electrolyte interface. It was found that the primary variation in the H-bond environment of interfacial Li^+ -bound water molecules could be classified by their H-bond donor number (N_{donor}), which ranges from 0, 1, and 2, corresponding to three types of water observed by Raman spectra. Meanwhile, we found that the proportion of interfacial water molecules with $N_{\text{donor}} = 2$ decreases gradually, and the fractions of water molecules with $N_{\text{donor}} = 1$ and 0 increase correspondingly at more negative potentials. Therefore, we established a correlation between the interfacial Raman spectra and MD simulation based on the relationship between N_{donor} and ν_{OH} , that is the Peak 1, 2, and 3 (in the order of increasing wavenumbers) of the interfacial Raman spectra are attributed to interfacial Li^+ -bound water molecules with two ($N_{\text{donor}} = 2$), one ($N_{\text{donor}} = 1$), and zero ($N_{\text{donor}} = 0$) H-bonds, respectively. Although we agree with this reviewer that the OH stretch frequencies could also be associated with the type of H-bonds (e.g., OH...water and OH...TFSI⁻), we assume that the difference caused by the hydrogen bond partner is much smaller than the shift caused by the change in hydrogen bond donor numbers.

Revision:

On **page 7**, the above discussion is added in the manuscript as:

“Inspired by the assignments of Raman spectra of water in the low-concentrated aqueous electrolytes^{22,32,34-37}, the spectra can be well fitted by the sum of three Gaussian bands, Peak 1, 2, and 3 (with increasing frequencies), suggesting three major types of water molecules in the EDL.”

4) The authors made a statement (in Abstract and throughout the main text) that the entire SHINERS spectra can be fitted with three bands. They justified the validity of this fitting analysis by comparing

their MD simulation results. Again, because of the issue (2) above, such Raman spectral analyses are questionable.

Reply 4:

We thank the Reviewer for this comment. As discussed in the reply to Comment 3 of this Reviewer, we analyzed the Raman spectra interfacial water at the electrode-WiS electrolyte interface with three bands, inspired by previous work where the Raman spectra of interfacial water have been assigned to three distinct components (Refs: *Nat. Mater.* 18, 697–701, 2019; *Nature*, 600, 81–85, 2021). Then we performed MD simulations to investigate the structure of water molecules and the related H-bond network at the electrode-WiS electrolyte interface to help assign the observed interfacial OH stretching spectra. Based on the relationship between N_{donor} and ν_{OH} , we finally established a correlation between the interfacial Raman spectra and MD simulation that Peak 1, 2, and 3 (in the order of increasing wavenumbers) of the interfacial Raman spectra are attributed to interfacial Li^+ -bound water molecules with two ($N_{\text{donor}} = 2$), one ($N_{\text{donor}} = 1$), and zero ($N_{\text{donor}} = 0$) H-bonds, respectively. Thus, it is reasonable to fit the SHINERS spectra with three bands.

Revision:

On page 7, The above discussion is added in the manuscript as:

“Inspired by the assignments of Raman spectra of water in the low-concentrated aqueous electrolytes^{22,32,34-37}, the spectra can be well fitted by the sum of three Gaussian bands, Peak 1, 2, and 3 (with increasing frequencies), suggesting three major types of water molecules in the EDL”

5) The main finding in this paper is the notable change of the OH peak frequency as the electric potential changes from 0 to -1.55 V (Fig. 2b). The E-field amplitude calculated from the MD simulations also shows the same trend. The authors jumped to the conclusion that the local solution structures in the vicinity of the Au surface obtained from the MD simulations faithfully reflect those in the region confined by Au NP and Au surface. But, again, their MD ignored the effect of induced polarization of Au NP on the spatially confined electrolyte solution.

Reply 5:

Thanks for the comment of the Reviewer. In the study of electrical double layer using SHINERS method, the computational methods have been properly used to correlate the experimental Raman

observations, for instance, *ab initio* molecular dynamics (AIMD) simulations without Au NP used for the SHINERS studies of interfacial water structures at Au(111) electrode (Ref: *Nat. Mater.* 18, 697–701, 2019) and Pd(*hkl*) electrode (Ref: *Nature*, 600, 81–85, 2021).

As discussed in the reply to Comment 1 of this Reviewer, the perturbation caused by the SHINs on the EDL structure on the Au electrode can be neglected. Thus, the experimental Raman spectroscopic results can be compared with the MD simulations.

Minor comment 1:

In Fig. S4, three Raman spectra are shown. What do the authors mean by “bulk phase”? Also, it is not clear why the baseline increases with decreasing frequency (Raman shift).

Reply:

Thank the reviewer for carefully reading. The “bulk phase” means the bulk electrolyte. As shown in the schematics in Fig. R11a, in the Raman spectroscopic measurement of bulk phase of WiS electrolytes (i), the laser beam penetrates into electrolyte bulk without the presence of electrode. Differently, the laser beam is focused onto the electrode surface in the measurement of interfacial region (ii).

Figure R11. (a) Schematics of Raman spectroscopic measurements of bulk phase and interfacial region. (b) UV-vis absorbance of 21 m LiTFSI electrolyte with H₂O as reference. The excitation wavelength of Raman spectroscopic experiment at 633 nm is marked by the dotted line.

The baseline is due to the weak fluorescence background from the electrolyte bulk, as indicated by the weak absorbance of 21 m LiTFSI electrolyte at 633 nm (the wavelength of excitation used in Raman spectroscopy) in the UV-vis absorbance data (Fig. R11b). This small baseline is in line with the Raman spectrum of concentrated LiTFSI electrolyte in literature (Ref: *Energy Storage Mater.* 38, 454-461, 2021). However, the small background can be cancelled out by the subtraction method used in this work.

Revision:

The schematics of Raman spectroscopic measurements of bulk phase and interfacial region were added in the updated **Supplementary Fig. 5** for clarification.

Supplementary Fig. 5 | Raman spectra of the O-H stretching mode at a Au(111) surface measured in 21 m LiTFSI electrolyte at 0.5 and 0.9 V, respectively. Raman spectrum from bulk phase is presented for comparison. The corresponding schematics for Raman spectroscopic measurements of bulk phase and interfacial region are shown in the right.

Minor comment 2:

In the first paragraph on page 9, the text refers to Supplementary Figs. 7 and 8, the captions of which say ‘simulated accumulative number densities’. Here, the meaning of ‘accumulative’ is ambiguous. What is the difference between ‘accumulated number density’ and just ‘number density’? Also, are

there any quantitative criteria to define the ‘interfacial region’?

Reply:

Thanks for this careful comment. In the manuscript, the number density (ρ_n) is defined as the number per volume for the center-of-mass of each species, with the unit: # nm⁻³. As for the ‘simulated accumulative number density’ in the manuscript, it is defined as the cumulated number of water or ions per unit electrode area in the interfacial region: $\rho = \int_0^{d_1} \rho_n dz$, where ρ_n is the number density as a function of distance from the electrode surface, and d_1 is the specified distance of the ‘interfacial region’. In terms of the ‘interfacial region’, it is the first adsorbed layer in this manuscript. Specifically, we have considered the first adsorbed layer as 0-0.4 nm for H₂O.

Revision:

The definitions of the accumulative number density and interfacial region have been added in the caption of **Supplementary Figs. 10 and 11**, as:

“The accumulative number density is defined as the cumulated number of Li⁺-bound water per unit area in the interfacial region, where the interfacial region is the first adsorbed layer (0-0.4 nm for Li⁺-bound water).”

Minor comment 3:

In Figs. 3e and 3f, the authors assign the peak near $\theta_{\text{Dipole}} = 60$ degrees as ‘H-up’ configuration. However, a water molecule satisfying the condition $\theta_{\text{Dipole}} = 60$ can have another rotational degree of freedom with the rotation axis along the direction of the water dipole so that the water plane makes 60 degrees with respect to the surface normal, in which case the OH bonds are rather far from the up direction. A kind of two-dimensional plot employing both θ_{Normal} and θ_{Dipole} would be more helpful to see the importance of the ‘H-up’ configuration.

Reply:

Thanks for the Reviewer’s comment. It is indeed that a water molecule with a specific dipole arrangement can have another rotational degree of freedom (i.e., the normal arrangement). As shown in Fig. R12, as for the water molecule with $\theta_{\text{Dipole}} = 60^\circ$ and $\theta_{\text{Normal}} = 60^\circ$, the OH bonds

point to the up direction. Meanwhile, when the θ_{Normal} turns to 90° (one of the major configurations under high polarization) or $\sim 20^\circ$ (indicating the normal orientation of water molecule near parallel to the electrode surface), the OH bonds still point in the up direction. Thus, it is reasonable to determine the ‘H-up’ configuration with the θ_{Dipole} peaking at around 60° . It is a good suggestion to determine the ‘H-up’ configuration by both θ_{Normal} and θ_{Dipole} . As suggested by the Reviewer’s Minor comment 4, each water molecule owns two OH bonds. Thus, it is meaningful to define the ‘H-up’ configuration with the OH bond. Therefore, following the Reviewer’s Minor comment 4, we analyze the ‘H-up’ configuration with OH bond. Details can be referred to the Reply for the Minor Comment 4.

Figure R12. Schematic for the arrangement of water with $\theta_{Dipole} = 60^\circ$ and various θ_{Normal} , i.e., $\sim 20^\circ$ (a), $\sim 60^\circ$ (b), and $\sim 90^\circ$ (c).

Minor comment 4:

The last paragraph on page 14 with Fig. 4 gives the argument that under low polarization, the interfacial water molecules adopt parallel and H-down structures, and under high polarization, $\sim 30\%$ of the Li-bound water adopts ‘H-up’ configuration, and the remaining Li-bound water molecules are in ‘H-down’ structure. I think this is not fully consistent with the results shown in Fig. 3b. The ‘H-down’ structure population is quite small under low polarization. The most conspicuous change in Fig. 3b with increasing polarization is the increase of the peak in the first layer, and the change in the third layer is quite minor. Also, I note that the strongest peak under high polarization in Fig. 3b is still the second peak corresponding to the parallel OH bond. The plot in Fig. 3b shows that a significant number of parallel OH bonds rotates to give the ‘H-down’ OH bonds as the polarization increases, while a small number of parallel OH bonds rotates to give the ‘H-up’ OH bonds. A single water molecule has two OH bonds, and a single water molecule can have a parallel OH and an ‘H-up’ or ‘H-down’ OH bond. Therefore, the estimated percentages of parallel, H-up, and H-down OH bonds would be informative, along with the percentages of ‘H-up’ and ‘H-down’ water molecules. Also, the authors’ argument seems to imply that the Li-bound water molecules in a parallel structure are nearly non-existent under high polarization, which needs to be more explicitly stated if that is the observation from the MD simulation.

Reply:

We appreciate this constructive comment for the analysis of OH bonds. Each water molecule has two OH bonds. In order to better describe the arrangement of Li⁺-bound water molecules, we calculate the 2D angular distribution of two OH bonds of Li⁺-bound water as a function of applied potential (Fig. R13).

Figure R13. **a**, Schematic for the angle between the OH bond of the water and the normal direction of electrode surface (θ_{OH}). **b**, 2D angular distribution of two OH bonds of Li-bound water as a function of applied potential.

As suggested by the Reviewer, each OH bond can be classified into the ‘H-up’, parallel, or ‘H-down’. Firstly, we give a criterion to determine the ‘H-up’, parallel, and ‘H-down’ OH bond. Since the angular distribution in Fig. R13 is a solid angle, for a bulk water system where water is randomly distributed, as shown in Fig. R14a, the angular distribution can be obtained as:

$$P(\theta)d\theta = \frac{S_{d\theta}}{S_{total}} = \frac{2\pi r^2 \sin \theta d\theta}{4\pi r^2} = \frac{1}{2} \sin \theta d\theta$$
$$P(\theta) = \frac{1}{2} \sin \theta$$

Thus, the angular distribution in a bulk system obeys the *sine* function distribution (Fig. R15b). To equally assign these three configurations (Fig. R14b), *i.e.*, $P(H - up) = P(parallel) = P(H - down) = 1/3$, the angle between the OH bond of the water and the normal direction of the electrode surface (θ_{OH}) can be classified into three regions: 0~70°, 70~110°, and 110~180° for ‘H-up’, parallel, and ‘H-down’ OH bond, respectively. With this criterion, a water molecule can be described as ‘H-up’, parallel, and ‘H-down’. In order to better describe the transition of the arrangement of Li⁺-bound water

molecules with the applied potential, we then calculate the differential 2D angular distribution of two OH-groups of Li^+ -bound water relative to the arrangement of water under PZC. As shown in Fig. R15, it can be seen that under low polarization, the arrangement of Li^+ -bound water molecules adjusts from parallel to ‘H-down’. Nevertheless, under high polarization, though the major Li^+ -bound water molecules transfer from parallel to ‘H-down’, we also notice that part of the water molecules go into the H-up configuration.

Figure R14. **a.** Schematic for the solid angle. **b.** Angular distribution for a homogeneous system. The angle between the OH bond of the water and the electrode surface (θ_{OH}) can be classified into three regions: $0\sim 70^\circ$, $70\sim 110^\circ$, and $110\sim 180^\circ$ for ‘H-up’, parallel, and ‘H-down’ OH bond, respectively.

Figure R15. Differential 2D angular distribution of two OH-groups of Li^+ -bound water relative to the arrangement of water under PZC.

To describe the transition quantitatively, the percentage of Li^+ -bound water with different configurations is shown in Fig. R16. It can be seen that the amount of interfacial Li^+ -bound water adopting a ‘H-up’ orientation increases from ~10% (PZC) to ~20% (high polarization); meanwhile, the amount of interfacial Li^+ -bound water with a parallel orientation decreases from ~43% to ~16%.

Figure R16. The percentage of Li^+ -bound water with different configurations.

Revision:

Following the Reviewer’s comment, the above discussion has been incorporated into the revised manuscript as:

“In order to describe the structure transition of interfacial Li^+ -bound water molecules with the applied potential more accurately, we then calculated the differential 2D angular distribution of two OH-groups of Li^+ -bound water relative to the arrangement of water under PZC. Specifically, the OH bond of water molecule can be classified into ‘H-up’, parallel, and ‘H-down’ with the angle between the OH bond of water and the normal of the electrode surface ranging in 0~70°, 70~110°, and 110~180°, respectively (Supplementary Fig. 12). As shown in Fig. 3f, the arrangement of Li^+ -bound water molecules adjusts from parallel to ‘H-down’ under low polarization. Nevertheless, under high polarization, though the

major Li^+ -bound water molecules transfer from parallel to ‘H-down’, part of the water molecules transfer into an unusual H-up configuration.” (Pages 13-14)

“As a result, the amount of interfacial Li^+ -bound water adopting a ‘H-up’ orientation increases from ~10% (PZC) to ~20% (high polarization); meanwhile, the amount of interfacial Li^+ -bound water with a parallel orientation decreases from ~43% to ~16% (Supplementary Fig. 16).” (Page 16)

The differential 2D angular distribution of two OH-groups of Li^+ -bound water relative to the arrangement of water under PZC (Fig. R15) is incorporated into the revised **Figure 3f**.

Fig. R14 and Fig. R16 are added in the **updated Supporting Information** as **Supplementary Figs. 12 and 16**.

Minor comment 5:

I think one missing information regarding the ‘H-up’ structure is what percentage of the ‘H-up’ OH bonds are involved in hydrogen bonding and what percentage of them are dangling OH bonds. In addition, the authors say on page 14 that ~30% of the Li-bound water molecules are in an ‘H-up’ structure under high polarization. But, a more significant quantity would be the difference in the percentage of the ‘H-up’ configuration between low and high polarization. So, the percentage of ‘H-up’ structure in low polarization also needs to be given for comparison, although the change of the 3rd peak in Figure 3b provides qualitative information.

Reply:

Thanks for this useful comment. The potential-dependent probability of interfacial ‘H-up’ Li^+ -bound water with different donor numbers has been shown in Fig. R17. In terms of the ‘H-up’ structure of Li^+ -bound water, ~20% of interfacial Li^+ -bound water molecules with ‘H-up’ structure form zero H-bond (i.e., two dangling OH bonds), and ~55% of interfacial Li^+ -bound water molecules with ‘H-up’ structure adopt one H-bond (i.e., one dangling OH bond); meanwhile, the rest ~25% of interfacial Li^+ -bound water molecules with ‘H-up’ structure could form two H-bonds (i.e., zero dangling OH bond). Additionally, as shown in Fig. R16, the amount of interfacial Li^+ -bound water adopting a ‘H-up’ orientation increases from ~10% (PZC) to ~20% (high polarization).

Figure R17. Potential-dependent probability (%) of interfacial ‘H-up’ Li^+ -bound water with different donor numbers.

Revision:

On **Page 16**, following the Reviewer’s comment, the above discussion has been incorporated into the revised manuscript as:

“As a result, the amount of interfacial Li^+ -bound water adopting a ‘H-up’ orientation increases from ~10% (PZC) to ~20% (high polarization); meanwhile, the amount of interfacial Li^+ -bound water with a parallel orientation decreases from ~43% to ~16% (Supplementary Fig. 16). Specifically, in terms of the ‘H-up’ structure of Li^+ -bound water, ~20% of interfacial Li^+ -bound water molecules form zero H-bond (*i.e.*, two dangling OH bonds), and ~55% of interfacial Li^+ -bound water molecules adopt one H-bond (*i.e.*, one dangling OH bond); meanwhile, the rest ~25% of interfacial Li^+ -bound water molecules form two H-bonds (*i.e.*, zero dangling OH bond, Supplementary Fig. 17).”

The Fig. R17 is provided in the **updated Supplementary Information (Supplementary Fig. 17)** for clarification.

REVIEWER COMMENTS

Reviewer #1 (Remarks to the Author):

Additional information provided in the reply shows an onset of reduction at -0.8 V. This indicates an onset of electrolyte decomposition in the very important region that is the focus on this manuscript. Authors state, however, that there is no OH⁻ (LiOH), LiF or other species detected from the spectroscopic signature observed at this potential and no visible SEI formation. SEI formation was observed at more negative potentials, -1.5 V.

Thus, there is no explanation explanation in the manuscript for what happens at -0.8 V during electrolyte reduction. I am skeptical that one can ignore the reduction process at -0.8 V and proceed with analysis of spectra in this region as no electrochemical processes occur (only double layer changes).

As I pointed out in the original review, "The influence of SiO₂ on the interfacial structure was not discussed". It is still not addressed in the revision.

Nevertheless, manuscript provides new and interesting data that would be interesting to the community and would start scientific discussion.

Reviewer #2 (Remarks to the Author):

I appreciate the authors' careful consideration of my comments and I am now willing to recommend publication.

Reviewer #3 (Remarks to the Author):

I have examined all the authors' responses to my comments on the original manuscript, "Unconventional interfacial water structure of highly concentrated aqueous electrolytes at negative electrode polarizations". I appreciate the authors' efforts to address them.

In comment 1, I raised a question about whether the water structure (EDL) and intermolecular (water-water, water-Au electrode, water-ion, ion-Au electrode) interactions are not affected by (~60 nm) Au NP near the Au electrode. The experimental configuration for SHINERS with very large (compared to molecules in between) shell-isolated nanoparticles (SHINs) differs from the realistic surface of the electrochemical Au electrode without such large polarizable metallic nanoparticles. Of course, I know that (as pointed out by the authors again in their reply 1) it is difficult (impossible) to obtain the Raman signal of interfacial water on an atomically flat electrode surface without plasmonic enhancement. However, this is not an appropriate answer to the question. In response to this question, the authors referred to the previous works (Nature, 464, 392–395, 2010; Nat. Commun. 7, 12440, 2016; Nature, 600, 81–85, 2021), where they independently found no electron transfer between the SHINs and the Au electrode during the electrochemical process. The authors took two figures from these reference papers to show that the SHINs are electrically inert. Also, the authors mentioned that a few other articles previously published already showed that the SHINERS is proven to be useful for studying the adsorption of ions or molecules and electrochemical reactions. To address this issue, the authors carried out another experiment and showed that the Raman spectra of adsorbed TFSI⁻ and water do not depend on the surface coverage of SHINs. The authors said, "If the SHINs have an influence on the water (H-bonding) structure, ion-ion interaction, and ion-water interactions, the Raman spectra of anions and interfacial water will change with the increase of SHINs." As the number of SHINs on the surface, the amount of water molecules under the enhanced Raman scattering effect increases linearly. Although this additional experimental result (Figure R10) might not prove that the intermolecular interactions, thereby water orientation and structure in the region where the enhancement of Raman scattering is significant, are not affected by SHINs, I believe that the authors and previous workers have shown that the local electric field due to the induced polarization of SHINs at finite applied potential is not drastically different from that in the absence of SHINs. Thus, I agree with the authors on this point and find the manuscript suitably revised.

Another issue I raised in the review was whether the band assignment (spectral analysis) of measured SERS spectra is correct or not. It should be noted that the mole fractions of ions in their electrolyte solutions are quite high. The authors, however, used the same band assignments that were used for low-concentration aqueous solutions (Li et al., Nat. Mater. 18, 697–701, 2019) to interpret their SERS spectra for extraordinarily high-concentration aqueous solutions in the present work. It is not clear whether such a spectral analysis method is valid for such highly concentrated solutions where water molecules can form strong H-bonding interactions with anions. However, their MD simulations show

that the effective (local) concentration of TFSI⁻ anions at the Au electrode-water interface is low, suggesting that the effect of anions on the vibrational frequency shift of interfacial water molecules is weak. So, I find their approach reasonable, but still, there should be further experimental studies proving this assumption.

The authors have addressed the other minor issues adequately. Overall, I believe that the revised manuscript is now suitable for publication in Nature Communications.

Reviewer #1 (Remarks to the Author):

Comment 1:

Additional information provided in the reply shows an onset of reduction at -0.8 V. This indicates an onset of electrolyte decomposition in the very important region that is the focus on this manuscript. Authors state, however, that there is no OH⁻ (LiOH), LiF or other species detected from the spectroscopic signature observed at this potential and no visible SEI formation. SEI formation was observed at more negative potentials, -1.5 V. Thus, there is no explanation explanation in the manuscript for what happens at -0.8 V during electrolyte reduction. I am skeptical that one can ignore the reduction process at -0.8 V and proceed with analysis of spectra in this region as no electrochemical processes occur (only double layer changes).

Reply:

We thank the Reviewer for this insightful comment. As shown by the CV of Au(111) in 21 m water-in-salt electrolyte (Supplementary Fig. 3b), a cathodic current starts at ~ -0.8 V, which can be explained by the reduction of H₂O molecule and TFSI⁻ anion (Ref: *Science* 350, 938-943, 2015).

However, although the reduction of electrolyte has started at -0.8 V, no SEI film formation was observed on the electrode surface as characterized by the *ex situ* AFM and Raman spectroscopic experiments (Supplementary Figs. 8 and 9). Hence, before the SEI formation at very negative potential (negative than ~ -1.55 V), the potential-dependent shift of ν_{OH} mode of interfacial water is still dominated by the Stark effect. As shown by the *in situ* Raman spectra in Fig. 2a in the manuscript, the Raman frequencies of Peak 1-3 decrease linearly from ~ 0 to -1.15 V, consistent with Stark effect induced frequency shifts (Refs: *Acc. Chem. Res.* 48, 998-1006, 2015; *J. Chem. Soc., Faraday Trans.* 92, 3829-3838, 1996), caused by the monotonically increasing total electric field experienced by interfacial water in the potential region of 0 to -0.96 V, which is in a good agreement with our MD simulation in Fig. 2d in the manuscript.

Revision:

On **page 9**, to explain the onset of the reduction wave at -0.8 V in the CV, the discussion is added:

“In addition, as shown by the CV of Au(111) in 21 m WiS electrolyte (Supplementary Fig. 3), a cathodic current starts at ~ -0.8 V, which can be attributed to the hydrogen evolution reaction and also

the reduction of TFSI⁻⁸. To characterize the surface change,...As shown in Supplementary Fig. 8, there is no observable Raman signal of SEI (*i.e.*, the deposition of LiOH) on the electrode surface under such potentials. Furthermore, the atomic force microscope (AFM) measurements were performed to study the surface morphology of electrode surface after holding at about -1.2 and -1.8 V.”

Comment 2:

As I pointed out in the original review, "The influence of SiO₂ on the interfacial structure was not discussed". It is still not addressed in the revision.

Reply:

Thank the Reviewer for the comment. First of all, in the SHINERS method we used to probe the interfacial layer, the ultrathin silica shell of SHINs is electrically and chemically inert, which can prevent the electron transfer between the Au nanoparticle and the electrode surface under bias conditions. This inert property of silica shell has been proven by several groups independently (Refs: *Nature*, 464, 392–395, 2010; *J. Raman Spectrosc.* 43, 46-50, 2012; *Nat. Commun.* 7, 12440, 2016; *Nat. Commun.* 12, 3264, 2021; *Nature*, 600, 81–85, 2021). Furthermore, the literature suggests that the SHINERS method have been proven to be suitable for the investigations of electrochemical interfaces, such as specific adsorption of ions (Refs: *Nature*, 464, 392–395, 2010; *J. Am. Chem. Soc.*, 142, 9439-9446, 2020, *J. Raman Spectrosc.* 43, 46-50, 2012; *Nat. Commun.* 12, 3264, 2021) and the **structure of interfacial water and electrical double layer** (Refs: *Nat. Mater.* 18, 697-701, 2019; *Nature*, 600, 81–85, 2021).

For instance, it has been reported that the electrochemically inert SiO₂ shell can protect the SHINs from oxidation and reduction during the potential scan, as indicated by the CVs of glassy carbon electrode covered with SHINs (Ref: Fig. S11 in *Nature*, 464, 392–395, 2010). Meanwhile, as shown in the previous report (Ref: Fig. S1 in *Nat. Commun.*, 7, 12440, 2016), the CV of Pt(111) electrode covered by SHINs shows the similar features to that of a bare Pt(111) electrode in 0.1 mol/L HClO₄. In particular, the characteristic features of hydrogen adsorption/desorption and the phase transition of OH on SHINs covered Pt(111) electrode are similar with that on bare Pt(111) electrode. All of the CVs

of SHINs covered glassy carbon electrode and Pt(111) electrode revealed the inert surface chemical property of the thin SiO₂ shell.

Importantly, in “Response to referees” file of the first revision, to explore the influence of SHINs upon the interfacial structures, we carried out a control experiment to compare the spectroscopic features obtained with **low and high** SHINs coverages on the Au(111) electrode surface. As shown in Fig. R1a, ~11% and ~28% coverages of SHINs were used for this comparison. If the SHINs (*i.e.*, SiO₂ shell) have an influence upon the interfacial structures, such as the structures of interfacial water, the Raman peak of ν_{OH} mode of interfacial water will change with different coverage of SHINs. As indicated by the Raman spectra shown in Figs. R1b and c (the same figures as Fig. R10 in “Response to referees” file of the first revision), the shapes and frequencies of TFSI⁻ and interfacial water are not influenced with varying SHINs coverage. Only the intensity changes, as different amounts of SHINs on the surface provide different Raman signal enhancements. Meanwhile, we used a subtraction method to remove the potential-independent contributions of Raman signals from the SHINs particle and bulk electrolyte. Therefore, the impact of the SiO₂ shell of SHINs upon the interfacial structures could be negligible.

At the same time, as pointed out by the Reviewer #3 in Comment 1 of this round of review (see below), and we quote that, “*I believe that the authors and previous workers have shown that the local electric field due to the induced polarization of SHINs at finite applied potential is not drastically different from that in the absence of SHINs.*” Hence, the SHINERS method is suitable to study the structures of the interfacial layer in the electrochemical environment.

Figure R1. (a) The schematics of the SHINERS method using different SHINs coverage by dropping

different volume of as-prepared SHINs solution onto the electrode surface. The surface coverages of SHINs are calculated to be ~ 11% and ~28% for low and high coverage. (b) and (c). Raman spectra of adsorbed TFSI⁻ (b) and interfacial water (c) at Au(111) surface at -0.55 V vs. PZC with the low (black) and high (red) SHINs coverages. These figures are the same as Fig. R10 in the first-version “Response to referees” file.

Revision:

On **page 4**, to address the feasibility of the SHINERS method in spectroelectrochemistry and the inert property of the silica shell of SHINs, the discussion is added:

“As illustrated in Fig. 1a, to study the EDL structure, an electrochemical SHINERS method was used (details in Supplementary Fig. 1 and Supplementary Information), which has been proven to be suitable for the investigations at the electrochemical interface, ...”

“The thin SiO₂ layer (2 nm in thickness) is electrochemically inert, which can insulate the Au particles (~57 nm in diameter) from the single crystal Au electrode while still provides enhancement of the electric field”

Comment:

Nevertheless, manuscript provides new and interesting data that would be interesting to the community and would start scientific discussion.

Reply :

Thanks very much for the Reviewer’s evaluation. We greatly appreciate the Reviewer for taking the time to evaluate the manuscript and help us to improve our work.

Reviewer #2 (Remarks to the Author):

General Comment:

I appreciate the authors' careful consideration of my comments and I am now willing to recommend publication.

General Reply:

We thank the Reviewer very much for the appreciation and the high regard for our work. We are also grateful for the Reviewer's insightful comments in the last review, which significantly improved our manuscript and the interpretation.

Reviewer #3 (Remarks to the Author):

General Comment:

I have examined all the authors' responses to my comments on the original manuscript, "Unconventional interfacial water structure of highly concentrated aqueous electrolytes at negative electrode polarizations". I appreciate the authors' efforts to address them.

General Reply:

We appreciate that the Reviewer re-evaluated our work and thank the Reviewer very much for her/his approval of our work.

Comment 1:

In comment 1, I raised a question about whether the water structure (EDL) and intermolecular (water-water, water-Au electrode, water-ion, ion-Au electrode) interactions are not affected by (~60 nm) Au NP near the Au electrode. The experimental configuration for SHINERS with very large (compared to molecules in between) shell-isolated nanoparticles (SHINs) differs from the realistic surface of the electrochemical Au electrode without such large polarizable metallic nanoparticles. Of course, I know that (as pointed out by the authors again in their reply 1) it is difficult (impossible) to obtain the Raman signal of interfacial water on an atomically flat electrode surface without plasmonic enhancement. However, this is not an appropriate answer to the question. In response to this question, the authors referred to the previous works (Nature, 464, 392–395, 2010; Nat. Commun. 7, 12440, 2016; Nature, 600, 81–85, 2021), where they independently found no electron transfer between the SHINs and the Au electrode during the electrochemical process. The authors took two figures from these reference papers to show that the SHINs are electrically inert. Also, the authors mentioned that a few other articles previously published already showed that the SHINERS is proven to be useful for studying the adsorption of ions or molecules and electrochemical reactions. To address this issue, the authors carried out another experiment and showed that the Raman spectra of adsorbed TFSI- and water do not depend on the surface coverage of SHINs. The authors said, "If the SHINs have an influence on the water (H-bonding) structure, ion-ion interaction, and ion-water interactions, the Raman spectra of anions and interfacial water will change with the increase of SHINs." As the number of SHINs on the surface, the amount of water molecules under the enhanced Raman scattering effect increases linearly. Although this additional experimental result (Figure R10) might not prove that the intermolecular interactions, thereby water orientation and structure in the region where the enhancement of Raman scattering is significant, are not affected by SHINs, I believe that the authors and previous workers have shown that the local electric field due to the induced polarization of SHINs at finite applied potential is not drastically different from that in the absence of SHINs. Thus, I agree with the authors on this point and find the manuscript suitably revised.

Reply:

We appreciate the Reviewer for her/his time in evaluating our work and this insightful comment upon

the SHINERS method.

The plasmonic enhancement with the shell-isolated nanoparticles (SHINs) is required to obtain the Raman signal of interfacial water on an atomically flat electrode owing to the intrinsic limitation. As agreed by this reviewer, several previous works have independently found that SHINs are electrically inert and no electron transfer between the SHINs and the Au electrode during the electrochemical process; SHINERS is proven to be useful for studying the adsorption of ions or molecules and electrochemical reactions.

Comment 2:

Another issue I raised in the review was whether the band assignment (spectral analysis) of measured SERS spectra is correct or not. It should be noted that the mole fractions of ions in their electrolyte solutions are quite high. The authors, however, used the same band assignments that were used for low-concentration aqueous solutions (Li et al., Nat. Mater. 18, 697–701, 2019) to interpret their SERS spectra for extraordinarily high-concentration aqueous solutions in the present work. It is not clear whether such a spectral analysis method is valid for such highly concentrated solutions where water molecules can form strong H-bonding interactions with anions. However, their MD simulations show that the effective (local) concentration of TFSI⁻ anions at the Au electrode-water interface is low, suggesting that the effect of anions on the vibrational frequency shift of interfacial water molecules is weak. So, I find their approach reasonable, but still, there should be further experimental studies proving this assumption.

Reply:

We appreciate the Reviewer for this constructive comment on the band assignment of measured SERS spectra. We agree with the reviewer that the OH stretch frequencies could be dependent on the type of H-bonds, *i.e.*, the anion may also play a role in the vibrational frequency shift of interfacial water molecules and whether the simple relationship between N_{donor} and ν_{OH} established in dilute electrolyte solutions can be applied in concentrated electrolytes needs further examination in future studies.

Revision:

On **page 11**, we add the follow sentence on the paragraph that discusses the spectral assignment.

“Although our result suggests that the well-established relationship between N_{donor} and ν_{OH} of water for dilute solutions can also be applied in concentrated electrolytes, this notion should be further

examined in future studies. ”

Comment:

The authors have addressed the other minor issues adequately. Overall, I believe that the revised manuscript is now suitable for publication in Nature Communications.

Reply:

We appreciate the Reviewer for this positive comment and recommendation to *Nature Communications*.